# Recurrent circuit dynamics underlie persistent activity in the macaque frontoparietal network

**Eric Hart\*, Alexander C Huk**

Center for Perceptual Systems, Department of Neuroscience, Department of Psychology, The University of Texas at Austin, Austin, United States

**Abstract** During delayed oculomotor response tasks, neurons in the lateral intraparietal area (LIP) and the frontal eye fields (FEF) exhibit persistent activity that reflects the active maintenance of behaviorally relevant information. Despite many computational models of the mechanisms of persistent activity, there is a lack of circuit-level data from the primate to inform the theories. To fill this gap, we simultaneously recorded ensembles of neurons in both LIP and FEF while macaques performed a memory-guided saccade task. A population encoding model revealed strong and symmetric long-timescale recurrent excitation between LIP and FEF. Unexpectedly, LIP exhibited stronger local functional connectivity than FEF, and many neurons in LIP had longer network and intrinsic timescales. The differences in connectivity could be explained by the strength of recurrent dynamics in attractor networks. These findings reveal reciprocal multi-area circuit dynamics in the frontoparietal network during persistent activity and lay the groundwork for quantitative comparisons to theoretical models.

## Introduction

Persistent neural activity is prevalent in many cortical and subcortical areas (*Andersen et al., 1987*; *Bruce and Goldberg, 1985*; *Constantinidis and Steinmetz, 1996*; *Fuster and Alexander, 1971*; *Fuster and Jervey, 1981*; *Bolkan et al., 2017*; *Ferraina et al., 2002*; *Guo et al., 2017*; *Inagaki et al., 2019*; *Mays and Sparks, 1980*; *Schmitt et al., 2017*), and has been studied extensively in the context of delayed oculomotor response tasks (*Funahashi et al., 1989*; *Gnadt and Andersen, 1988*; *Hikosaka and Wurtz, 1983*). In these tasks, a subject must remember the location of a visual stimulus and make a saccadic eye movement to that location after a delay. During this delay period, single neurons in many primate brain areas, such as the lateral intraparietal area (LIP) and frontal eye fields (FEF), continue to spike vigorously in absence of the stimulus, leading up to the behavioral response (*Balan and Ferrera, 2003*; *Clark et al., 2012*; *Funahashi et al., 1989*; *Gnadt and Andersen, 1988*). This neural activity bridges sensory inputs with subsequent motor outputs and is thought to reflect the active maintenance of information in working memory, a process that forms the basis of many complex adaptive behaviors (*Baddeley, 2012*).

Because persistent activity lasts far longer than the intrinsic cellular time constants of most neurons, theoretical models of persistent activity have focused on how a mix of circuit and cellular mechanisms might interact to transcend the timescale of individual neurons. Classically, these models rely on a core architecture that includes recurrent excitatory connectivity, balanced inhibition, and a subset of single neurons with long (but realistic) intrinsic time constants (*Amit and Brunel, 1997*; *Compte, 2006*; *Compte et al., 2000*; *Tegnér et al., 2002*; *Wang, 1999*; *Wang, 2001*). Although these models are remarkable for using biologically plausible elements to show how the brain could in theory exhibit persistent activity at an aggregate level, little is known about the detailed patterns of circuit-scale spiking activity in the primate brain areas that originally motivated these theories.

**\*For correspondence:**
ehart004@gmail.com

**Competing interests:** The authors declare that no competing interests exist.

Furthermore, population models have yet to reconcile the fact that real neurons with persistent activity are often highly dynamic, heterogeneous, and exhibit mixtures of sensory and motor response components (*Meister et al., 2013*; *Premereur et al., 2011*; *Rigotti et al., 2013*).

To fill these gaps, we conducted simultaneous recordings of neural ensembles in LIP and FEF during a memory-guided saccade task and characterized the trial-by-trial dynamics in this circuit with a multi-neuron generalized linear model (GLM). This task is often employed to evoke persistent responses in both LIP and FEF, and is routinely used to identify neurons with long timescale dynamics that are thought to play a role in sensory integration (*Chafee and Goldman-Rakic, 1998*; *Gold and Shadlen, 2007*). It also serves as the fundamental test case for modeling persistent activity in theoretical studies (*Murray et al., 2017a*; *Murray et al., 2017b*; *Spaak et al., 2017*). The population-level GLM allowed us to isolate the effects of each neuron's spiking activity on the activity of other simultaneously recorded neurons, while statistically parceling out the effects of shared task variables (such as visual responses to the flashed target, and motor responses related to the impending saccade). This analysis revealed remarkably strong and slow recurrent excitation both within and between areas. Interneuronal coupling between LIP and FEF was highly reciprocal, without a clear asymmetry between feedforward and feedback connections. Furthermore, the spiking activity of most neurons was driven by spikes from other neurons more than the cell's own recent spike history. In addition to slow recurrent excitation, inhibition with moderate timescales was also present, followed by more standard fast timescale excitation. The existence of corresponding inhibitory interactions lends support to the idea that inhibition may act as a stabilizing mechanism to curb runaway excitation.

Together, these results demonstrate the core aspects of theoretical models (i.e. slow recurrent excitation, balancing inhibition) while also providing quantitative estimates of the various timescales and magnitudes of these theorized components. However, we also found that there were stronger local functional interactions within LIP than FEF, and that many neurons in LIP had longer spike-history and coupling timescales. These aspects of our results run contrary to recent theoretical hypotheses about the relative roles of LIP and FEF (e.g. *Murray et al., 2017b*), but concur with previous findings that examine correlations between pairs of neurons locally in each area (*Katsuki et al., 2014*). By analyzing synthetic spike trains from a classic 'bump' attractor network used to model spatial working memory (*Compte et al., 2000*; *Gerstner et al., 2014*), we found that the differences in coupling between LIP and FEF could be explained by differences in the strength of recurrent connections in attractor networks. Interestingly, this analysis revealed that both areas could support persistent activity independently, but between-area interactions alone were insufficient to generate persistent activity (despite reflecting statistically-significant coupling). Thus, our results support and refine key components in models of persistent activity, and offer unexpected insights into how and where the brain implements circuit-based computations critical for linking sensation and action over time.

## Results

### Simultaneous recordings in LIP and FEF during memory-guided saccades

In order to test whether recurrent functional interactions in the frontoparietal network underlie persistent activity, we recorded ensembles of neurons (~5–20 cells per session) in areas LIP and FEF simultaneously (*Figure 1a*; 967 units, LIP: 407; FEF: 560; 7448 pairs). This allowed us to measure and compare the neuronal interactions both locally (within a cortical area) and over longer distances (between cortical areas). Macaques performed a classical memory-guided saccade task with two crucial refinements. First, the timing of all trial events including the delay period was variable (*Figure 1b*). Second, the location of the target was continuously jittered in space, thus the ensuing saccades also had variability in their timings and endpoints. This trial-by-trial spatiotemporal variability allowed us to fit a population encoding model (on ensembles of simultaneously-recorded neurons) at the level of single trials which, in turn, enabled us to disentangle the relative influences of various task events and network activity on the spiking activity of each recorded neuron. The saccade target was presented either within the constellation of receptive fields ('in-RF') of the neurons under study (in the hemifield contralateral relative to the recording sites), or diametrically opposed in the

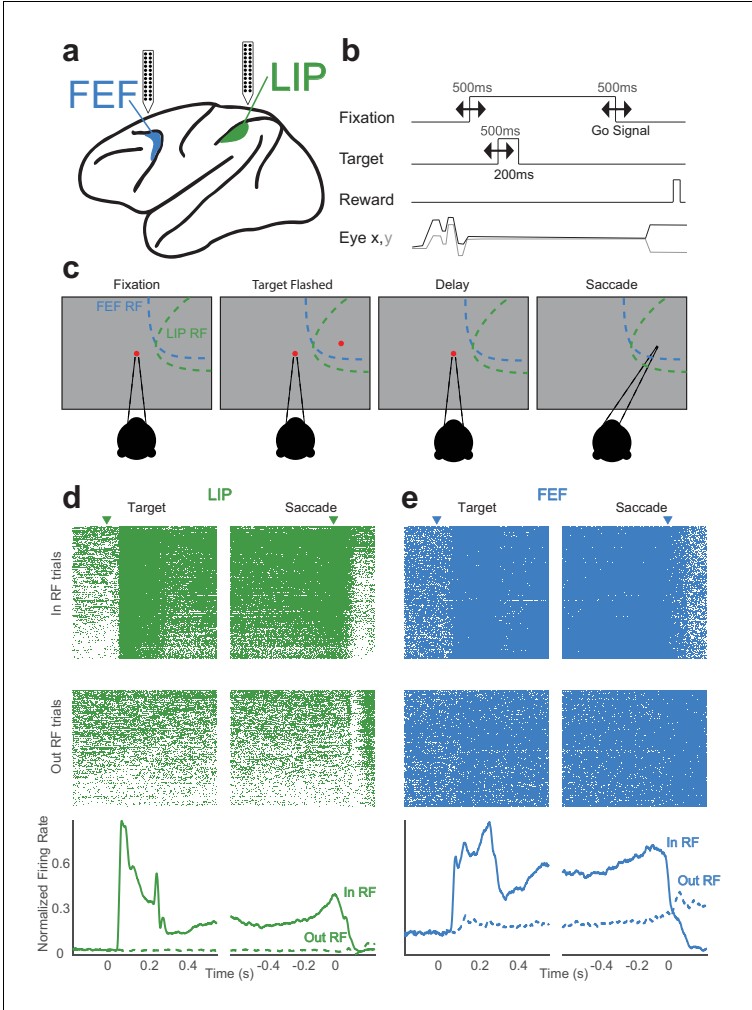

**Figure 1.** Memory-guided saccade task and simultaneously recorded neurons in LIP and FEF. (a) Linear (stereotrode) probes (24–32 channels) were used to record in both LIP and FEF. (b) The timings of task events were varied continuously from trial to trial, making the delay (memory) period variable. (c) The monkey fixates and a target is flashed in the periphery (either 'in' or 'out' of the receptive fields); after a delay period (in which the monkey must maintain fixation) the fixation point is removed, indicating a 'go-signal,' and the monkey must make a saccade to where the target was. (d,e) Simultaneously recorded neurons in LIP (d) and FEF (e) with canonical responses. Each has a transient response to the visual target and a sustained response throughout the delay period (PSTH, bottom image). Even in these example neurons with strong persistent activity, there is significant variability during single trials (rasters, top image).

ipsilateral hemifield ('out-RF'; *Figure 1c*). *Figure 1* (d,e bottom) shows two example simultaneously recorded neurons with canonical responses. Each example neuron responded transiently to the visual target, and exhibited stimulus-selective persistent activity throughout the delay period. However, even in these example cells (selected to show robust and fairly constant persistent activity), substantial variability was present in their single-trial responses, reinforcing the value of an analysis scheme that can capture single-trial dynamics when assessing interactions between neurons (rasters *Figure 1d,e* top).

Persistent activity is traditionally observed (and quantified) in the average responses of neurons across many trials, but during natural behavior, the monkey must remember the stimulus and make decisions based solely on individual trials. To characterize and exploit the dynamics of single-trial responses across the circuit, we developed a population encoding model (i.e. a fully coupled generalized linear model [GLM]; *Figure 2a*). This regression-based statistical approach describes the time-varying spike rate of each neuron in terms of the external task events (i.e. the visual target being

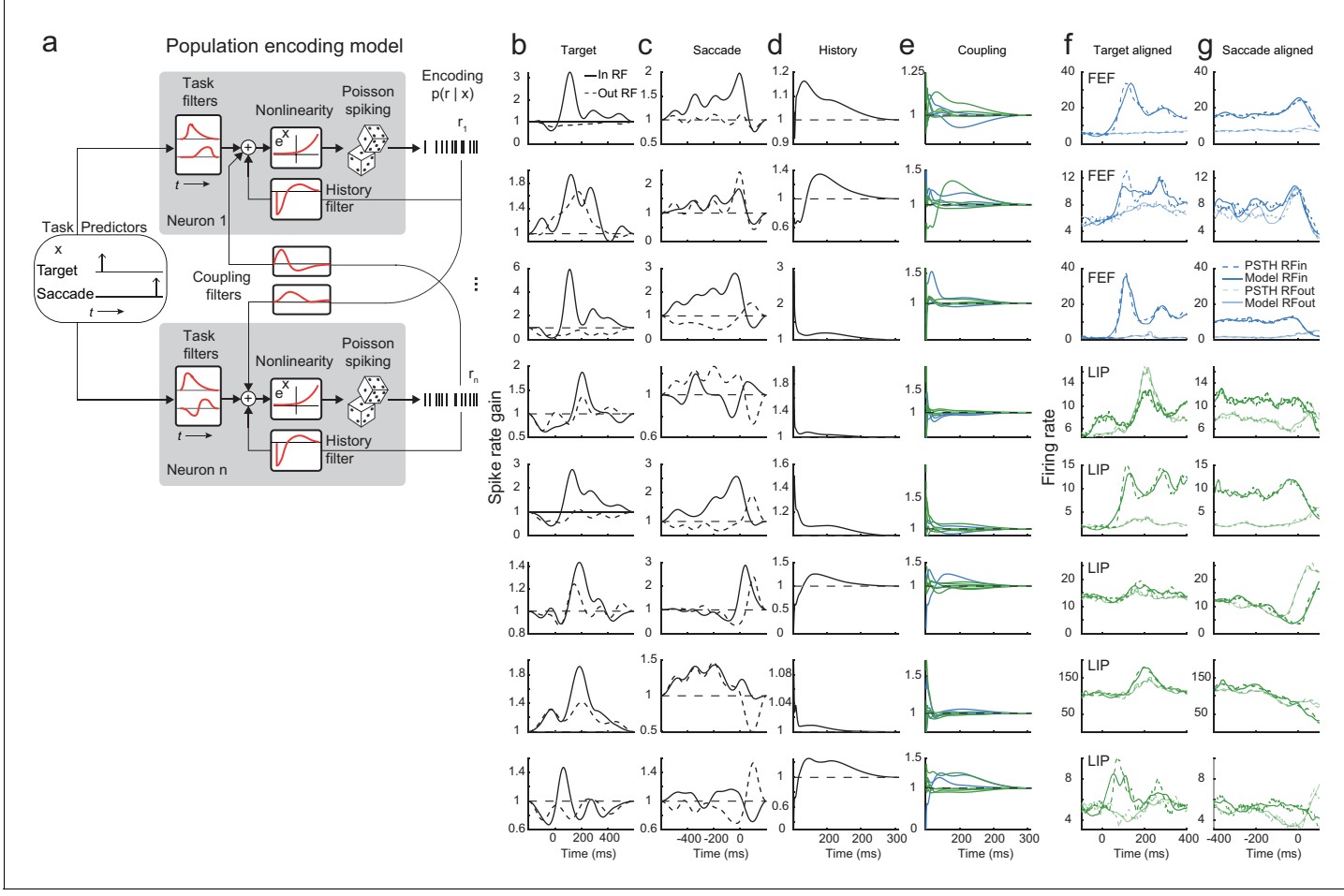

**Figure 2.** Population encoding model and analysis of example session. (**a**) This generalized linear model (GLM)-based approach describes the time-varying spike rate of each neuron in terms of the external task events (i.e. the saccade target being flashed and the saccadic response), the recent spike-history of that neuron, and the activity of other simultaneously recorded neurons (within and across regions). The model is defined by three stages: 1) a linear stage that takes a weighted sum of the covariates and temporal filters, 2) an exponential point non-linearity that converts the linear output to a spike rate, and 3) conditionally Poisson spiking. (**b–g**) Each row corresponds to a neuron (1–3 in FEF, 4–8 LIP) and each column corresponds to the kernel of a particular predictor: (**b**) onset of the visual target; target in RF (solid line) and out RF (dashed line), (**c**) time of the saccade response, (**d**) recent spike-history and (**e**) coupling from other neurons (in LIP: green; in FEF: blue). Each kernel represents the time-varying gain on spiking for the fitted neuron due to the particular predictor while simultaneously accounting for the influence of the other predictors. (**f,g**) Each column displays the true PSTH (dashed lines) and predicted PSTH (solid lines) for each neuron (rows) for targets in (dark lines) and out (light lines) of the RF, aligned to the (**f**) target on and the (**g**) saccade. The predicted PSTHs were constructed from generating single-trial spike trains from the full population model for each neuron and averaging them.

The online version of this article includes the following figure supplement(s) for figure 2:

**Figure supplement 1.** Cross-correlations of the example session (*Figure 2*) and the relationship between coupling and correlations.

flashed and the saccadic response), the recent spike-history of that neuron, and the activity of other simultaneously recorded neurons. This approach allowed us to investigate the single-trial interactions between neurons while parceling out the influences of the visual stimulus (*Figure 2b*, first column), the motor response (*Figure 2c*), and each neuron's autocorrelation (i.e. the 'spike-history'; *Figure 2d*). We characterized the functional interactions between neurons with 'coupling kernels,' which describe the statistical dependencies between single-trial spike trains over time (*Figure 2e*). Critically, these coupling terms explain variance not explained by the other predictors, and are fit directionally (i.e. they capture the impact of one neuron's spikes on the subsequent activity of another neuron). The population GLM goes beyond simple pairwise comparisons by fitting the model using spikes from all simultaneously recorded neurons. Thus, the 'full' population GLM

characterizes the response of the fitted neuron with respect to the spiking activity of all the other neurons, providing a less ambiguous characterization than standard one-pair-at-a-time pairwise correlations (*Stevenson et al., 2008*). This model-based approach provides a disentangled estimate of spike-on-spike coupling across all recorded neurons, while simultaneously parsing out the shared drive due to the stimulus or motor response (i.e. it naturally 'models out' the signal correlation).

The example session (*Figure 2b–e*) showcases the diversity of responses of eight simultaneously recorded neurons. The GLM captures all aspects of the apparent heterogeneity well: idiosyncrasies in temporal dynamics, variations in functional interactions, and cell-to-cell differences in stimulus dependence. As an initial standard validation, we simulated single-trial spike trains from the GLM fits and averaged these to calculate a predicted peristimulus time histogram (PSTH), which captured the average responses of each neuron quite well (*Figure 2f,g*). The GLM-derived measure of coupling reveals similar patterns as seen via traditional measures of correlations (*Figure 2—figure supplement 1*; *Pillow et al., 2008*) but offers the advantage that it can 'explain away' potentially misleading correlations by leveraging the simultaneous fits across the ensemble of neurons and thus provides a cleaner estimate of the true functional connectivity of the network (*Stevenson et al., 2008*). Although of course a perfect characterization would require measuring every single neuron in the network (*Aertsen et al., 1989*), these methods have been previously validated in physiological circuits with better-known connectivity (*Gerhard et al., 2013*; *Pillow et al., 2008*).

## Strong coupling exists both within and between areas but is stronger within LIP

Strong coupling was observed throughout the frontoparietal network. We quantified the overall coupling strength for each pair of simultaneously recorded neurons by summing the weight of each coupling kernel, allowing the overall effect of coupling for each neuron pair to be excitatory or inhibitory (i.e. one neuron was more or less likely to spike given another neuron's spike). *Figure 3a* (left column) summarizes the overall strengths of coupling for the four main classes of interneuronal interactions: both neurons within LIP ('LIP-LIP'), both neurons within FEF ('FEF-FEF'), from an LIP neuron to an FEF neuron ('LIP-FEF'), and from an FEF neuron to an LIP neuron ('FEF-LIP'). For all of these functional interactions, the observed coupling (colored bars in histograms) was far stronger than expected by a null distribution (gray bars, calculated by permuting the trial order of other neurons relative to the neuron being fitted, and then re-fitting the model; permuting the data in this way destroys the correlational structure between the neurons, but crucially preserves the spike times and model-based analysis of task events for all spike trains [*Pillow et al., 2008*]; Wilcoxon rank-sum test, LIP-LIP, $p=6.45\times10^{-79}$; FEF-FEF, $p=6.78\times10^{-165}$; LIP-FEF, $p=1.69\times10^{-86}$; FEF-LIP, $p=8.52\times10^{-84}$). A large fraction of the individual coupling kernels was substantially greater than the null distribution (>2 SD of mean of the null [~0]) for each area interaction (LIP-LIP = 60%; FEF-FEF = 54%; LIP-FEF = 60%; FEF-LIP = 58%).

Interneuronal coupling often manifested as excitatory interactions (i.e. positive values), but many instances of stronger-than-chance suppressive interactions were also evident (i.e. negative values). The coupling components were strongest for the within-LIP interactions, although within-FEF interactions were also far greater than chance levels. Between-area coupling (LIP-FEF and FEF-LIP) was also strong and spanned approximately the same range of magnitudes as the within-FEF interactions. These quantitative comparisons of the distributions of coupling strength are graphically apparent in the cumulative distributions (*Figure 3c*). The results also held for a comparison of the absolute coupling magnitudes, as well as a comparison of the overall excitation and inhibition (*Figure 3—figure supplement 1*).

Beyond assessing the overall magnitudes of each interaction, we examined the time courses of the coupling kernels by performing principal components analysis (PCA) on all of the kernels within each of the four interaction types (LIP-LIP, FEF-FEF, LIP-FEF, and FEF-LIP), and plotted the first three PCs (which together accounted for ~90% of the variance). Across all interaction types, coupling was dominated by long timescale excitation, indicating that spikes from one neuron increased the gain on spiking of other neurons for up to ~200 ms (*Figure 3b*, first PC/darkest curves). A second, sharper, fast excitation component followed (*Figure 3b*, 2nd PC/medium shaded curves), and was more pronounced within both areas than between areas, consistent with this component reflecting local interneuronal interactions. Finally, a third component was primarily suppressive (*Figure 3b*, 3rd PC/lightest curves), with a timescale between the first (slow) and second (fast) excitatory

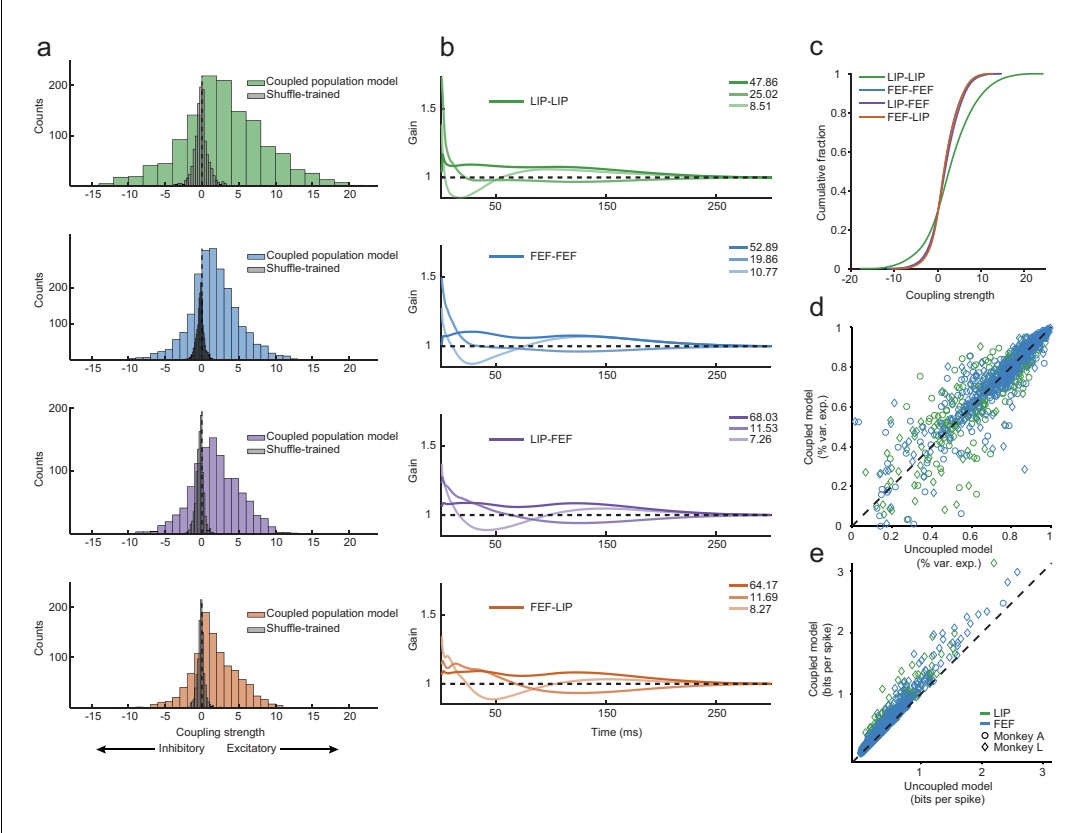

**Figure 3.** Exceptionally strong coupling both within and between areas. (a) Distributions of raw coupling weights (colored histograms) defined by the sum of each kernel for each area interaction (LIP-LIP, FEF-FEF, LIP-FEF, FEF-LIP). Positive weights denote an excitatory interaction (the fitted neuron was more likely to spike given that another neuron just spiked) and negative weights indicate an inhibitory interaction (the fitted neuron was less likely to spike given that another neuron just spiked). A null distribution was constructed by shuffling the order of the trials for each neuron (gray histograms), effectively destroying the correlations between the ensemble of neurons but preserving the stimulus dependence and spike times. These distributions describe the probability of observing coupling due to chance alone (Wilcoxon rank-sum test, $p=6.45\times10^{-79}$; $p=6.78\times10^{-165}$; $p=1.69\times10^{-86}$; $p=8.52\times10^{-84}$). (b) Principal components analysis was performed on the coupling kernels split by each area interaction (as in (a)); the first three principal components (dark to light lines) accounted for around 90% of the variance (legend: variance explained for each PC). (c) Cumulative fraction of coupling weights for each interaction, corresponding to the distributions in (a). (d) Variance explained of the predicted PSTHs of the coupled model vs. the uncoupled model (median 76% variance explained under both models). Coupling was unnecessary to capture the average response of most neurons (the points fell near the unity line). (e) Predicted single-trial spike times (in units of bits-per-spike) of the coupled model vs. the uncoupled model. Coupling improved the predictions of the single-trial spike trains by 22% on average (25% for LIP; 19% for FEF; points fell above the unity line). The online version of this article includes the following figure supplement(s) for figure 3:

**Figure supplement 1.** Absolute magnitudes of coupling and overall excitation and inhibition.

**Figure supplement 2.** Distributions of max coupling kernels (gain).

components. The suppressive component was proportional to the fast excitation component in magnitude and lag. These three timescales and signs of interaction are ubiquitous in biophysical models of persistent activity, and are posited to reflect the relative contributions of NMDA, AMPA, and GABA receptors, respectively (*Compte et al., 2000*; *Goldman-Rakic, 1995*; *Wang, 1999*).

We verified the fits across all sessions (*Figure 3d*) by comparing predicted PSTHs generated from the GLM to the true PSTHs (as in *Figure 2f,g*); the predicted PSTHs did a good job of explaining the real PSTHs of most neurons regardless of whether the neurons were treated independently (in an 'uncoupled' model with only task and history kernels) or not (in the 'full' coupled model with task, history, and coupling kernels; median 76% variance explained under both models). Although interneuronal coupling terms did not improve predictions of the average response (i.e., the PSTH; *Figure 3d*), they were important for capturing the spike times in single-trial spike trains (*Figure 3e*; the coupled model provides 22% more information on average). This distinction has been noted

previously (*Pillow et al., 2008*; *Runyan et al., 2017*) and means that the shared variability of the population carries information about a single neuron's spike trains beyond that provided by the task components alone, and is unsurprising given that the PSTHs naturally obscure the single-trial dynamics by averaging them away. These findings emphasize the value of this analysis scheme because it allows us to examine the single-trial dynamics of the network during the timescales relevant for working memory.

In addition to comparing the observed coupling to null (permutation-fitted) distributions and an uncoupled model (with independent neurons by design), we also compared our results in the LIP-FEF network to similar coupling quantifications from other brain areas and tasks. Coupling between simultaneously-recorded neurons in visual area MT (recorded in the lab for other projects), although detectable, was considerably weaker than observed in these experiments (mean gain of max coupling filters, MT-MT = 1.08; LIP-LIP = 1.34; FEF-FEF = 1.16, LIP-FEF = 1.10; FEF-LIP = 1.12; *Figure 3—figure supplement 2*). Likewise, analysis of coupling between MT and LIP during a perceptual decision-making task was unable to detect interactions beyond that expected by chance (*Yates et al., 2017*).

Thus, conventional statistical tests, quantitative and qualitative comparisons to a null distribution, and comparisons of effect sizes measured in other contexts, all point to the existence of exceptionally strong recurrent interactions within the frontoparietal network. These were most pronounced within area LIP but were prevalent throughout this multi-area circuit. These interactions were primarily excitatory, but also contained substantial instances of suppression. Between-area coupling was roughly symmetrical in strength and timescale (*Figure 3a,b*), consistent with anatomy (*Andersen et al., 1990*; *Ferraina et al., 2002*; *Selemon and Goldman-Rakic, 1988*). Together, these results suggest a reciprocal inter-area network rather than a serial multi-stage system.

## Interneuronal coupling outweighs spike-history dynamics, especially in LIP

In the prior section, we described how certain neurons impacted the probability of other neurons' subsequent spiking. It is also reasonable to expect that a cell's own spiking affects the probability of its subsequent firing. To characterize the relative impacts of (interneuronal) coupling and ('intrinsic') spike-history for each neuron, we defined predictive indices as the difference between the performance of the GLM with and without coupling, or with and without spike-history. For example, a coupling index of one would indicate that the neuron's spike trains are completely predictable from the spiking activity of other neurons, and a value of zero would mean that spikes from other neurons add no predictive power over and above the task predictors and the neuron's recent spike-history. The history index was constructed in an analogous fashion.

*Figure 4* shows the relative effects of interneuronal coupling and spike-history for neurons in both areas. A greater number of neurons in FEF were predicted by their own spike-history than in LIP (*Figure 4a*), whereas the spike trains of LIP neurons were better predicted by spikes from other neurons than the analogous case in FEF (*Figure 4b*). These contrasting results may reflect fundamental differences in the functional connectivity and distributions of cell types in the underlying neural populations (*Masse et al., 2017*; *Rosenbaum et al., 2017*). However, overall, neurons in both areas were more influenced by interneuronal coupling than by their own spike-history (*Figure 4c*), pointing to the importance of circuit-level interactions driving spiking activity. This is especially striking given that the coupling is estimated while also accounting for the responses of all neurons to the task events, and thus reflects the interactions while statistically controlling for shared responses due to visual and motor drive. Neurons in both LIP and FEF tended to be driven by either the concurrent spiking activity of the population, or by their own spike-history, with few neurons impacted equally by both (*Figure 4c*).

Across both areas, the ability to predict spikes for a large number of neurons was not improved by including a history term, meaning that their spike trains were predicted just as well by a model with only spikes from other neurons and the task components (*Figure 4c*). However, for a small subset of neurons (11%), the recent spike-history was far more predictive than coupling. Interestingly, these neurons were predominantly in FEF (nearly twice as many as in LIP; *Figure 4c*). Overall, these results imply that network dynamics play a more vital role than intrinsic dynamics for most neurons in both areas during persistent activity. Network contributions were especially strong in LIP, consistent with our prior observation that the magnitudes of the within-area coupling kernels were especially

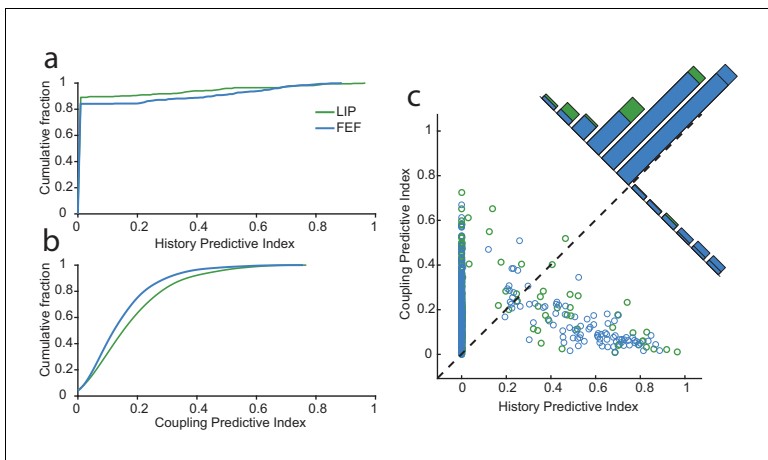

**Figure 4.** Network drive dominates circuit dynamics, especially in LIP. For each neuron, we constructed predictive indices to further assess the influence of the spiking activity from other neurons (coupling) and itself (history). These indices are defined as the difference between the deviance explained by the model with or without history (a) or with or without coupling (b). (a) The cumulative fraction of history indices for neurons in each area. Overall, more neurons in FEF than LIP were impacted by their recent spike-history. (b) Same as in (a) but for coupling. Conversely, more neurons in LIP were influenced by the spiking activity of other neurons. (c) Comparison of the history and coupling indices. On the whole, spiking activity of neurons in both regions was more predicted by coupling than history (88.4% of neurons). For a large fraction of neurons, the history term was uninformative over and above the coupling and task terms (many neurons had a history index of zero). For a small subset of neurons (11%), predominantly in FEF, history had far more influence than coupling.

The online version of this article includes the following figure supplement(s) for figure 4:

**Figure supplement 1.** The relationship between persistent activity, coupling, and spike-history.

large in LIP (*Figure 3a*). We also wanted to assess whether neurons with strong persistent activity (determined in the traditional way, via inspection of their PSTH) were more driven by coupling or spike-history effects. Interestingly, there was no clear relationship to either (*Figure 4—figure supplement 1*). This emphasizes the earlier point that the average responses of single neurons may reveal little about the single-trial dynamics of the network (*Latimer et al., 2015*; *Stokes and Spaak, 2016*). The most 'persistent' responses evident in trial-averaged assays might be those that are more reliably driven by regular task events (i.e. the pre-motor build-up or the decaying visual response, or both), as opposed to reflecting a unique and persistent network contribution.

The interplay and balance of intrinsic versus network time constants is a major component of successful models of persistent activity (*Cavanagh et al., 2018*; *Runyan et al., 2017*). We therefore assessed the relative contributions of spike-history and interneuronal coupling timescales. The history kernels are of course an intriguing but imperfect extracellular lens into the intrinsic time constants of the neurons, estimated by modeling the effect of recent spikes on that neuron's probability of spiking as a function of time and excluding the effects of task and coupling drive. Although the history kernel does not necessarily or directly describe intracellular or biophysical mechanisms, it does characterize the temporal response properties of the neuron due to 'self-drive' (e.g. long timescale autocorrelation, refractory periods, bursting dynamics etc.) as distinct from the effects of other (recorded) neurons and the task events that drive most (or all) neurons. We summarized the history kernels for each area with the first three principal components (*Figure 5a,b*). The dominant history component for both areas reflected a slow, gradually decaying time course. We estimated the spike-history time constants of each neuron by fitting an exponential curve to the history kernels (*Figure 5c*). Most neurons in both areas had short time constants, but the distributions had heavy right tails, with many neurons exhibiting longer timescales (100–300 ms). Overall, LIP had more neurons with long time constants than FEF (*Figure 5c*, inset), which may be surprising given the traditional notion that FEF lies higher up the cortical hierarchy of the visual system (which usually constitutes longer timescales; *Honey et al., 2012*; *Murray et al., 2014*). However, from the oculomotor point of view, the hierarchy is reversed, given that FEF receives direct input from subcortical

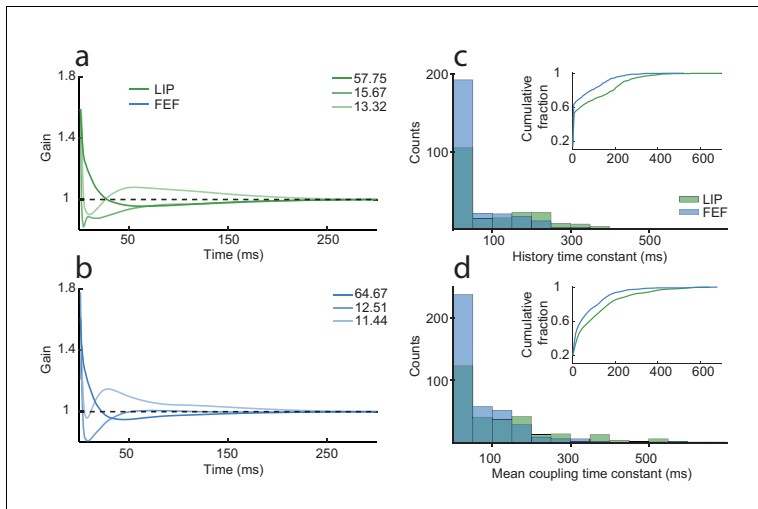

**Figure 5.** Spike-history and coupling time constants are similar but longer in LIP than FEF. The first three principal components for the history kernels in LIP (**a**) and FEF (**b**), which capture the spike-history time constant of the neurons, while controlling for the task and coupling drive. The dominant component in both areas was a slow decay. An exponential curve was fit to the history kernel of each neuron to estimate a time constant (tau). (**c**) The distribution of spike-history time constants differed between areas (Wilcoxon rank-sum test, p=$1.20 \times 10^{-06}$; LIP mean = 99.6 ms; FEF mean = 68.2 ms). Most neurons had short time constants, but there was a 'long tail' of neurons with much longer time constants (100–300 ms). (**d**) Same as (**c**) but for the mean coupling kernel of each neuron. Coupling time constants were larger overall than history time constants and also differed between areas (Wilcoxon rank-sum test, p=$6.43 \times 10^{-07}$; LIP mean = 133.8 ms; FEF mean = 76.5 ms). Insets shows the cumulative distributions of the time constants; LIP had more neurons than FEF with long coupling and history time constants.

areas involved in eye movements, and exhibits lower microstimulation thresholds for generating saccades (*Ferraina et al., 2002*; *Kimmel and Moore, 2007*).

We estimated the time constant from the mean coupling kernel for each neuron in an analogous way, and found that coupling time constants were similar to the spike-history time constants on the whole (*Figure 5d*). These findings are consistent with the idea that persistent activity results from a combination of network and intrinsic dynamics, both with relatively slow time constants that can extend into the 200–300 millisecond-long range. We emphasize that the population GLM estimates both the coupling and spike-history kernels while simultaneously accounting for the effects of the task variables that presumably drive most, if not all, of the neurons. In simpler, traditional analyses (such as the autocorrelation from binned spike counts), the impact of visual and motor drive would overshadow and confound the measured network dynamics (*Cohen and Kohn, 2011*).

## Interneuronal coupling is dynamic across behavioral epochs

The nature of neural population dynamics during working memory is currently a matter of debate (*Cavanagh et al., 2018*; *Constantinidis et al., 2018*; *Lundqvist et al., 2018*). Some models of working memory propose that the activity of the network should remain stable over time and others suggest that it can drift, or that the time-varying activity is critical to the computation (*Barak and Tsodyks, 2014*; *Stokes, 2015*). To examine how the dynamics of the network evolved over the course of the trial in the neural data, we compared coupling between several temporal epochs. As a baseline, we measured the coupling during the fixation period at the beginning of each trial and compared it to coupling during the delay period. In both cases, the visual input was identical (i.e. a fixation point and nothing else on the screen), but during the delay period the monkey had to remember the location of the target in order to perform the subsequent memory-guided saccade.

We found strong coupling across the circuit even during fixation, in accordance with earlier findings investigating local pairwise correlations (*Katsuki et al., 2014*). Rather than a profound global change in network interactions from fixation to delay, we observed a subtle yet systematic shift in increased excitatory coupling in all area interactions, with the most pronounced change occurring

locally within LIP (*Figure 6a*; Wilcoxon rank-sum test, LIP-LIP, p=4.87×10$^{-09}$; FEF-FEF, p=5.17×10$^{-04}$; LIP-FEF, p=1.64×10$^{-04}$; FEF-LIP, p=3.01×10$^{-03}$). This highlights the potential role of nuanced changes in the dynamics of the network, rather than an indiscriminate increase in overall population activity. This is consistent with work showing that the distribution of firing rates of the population in PFC remains relatively constant between the fixation period and the delay period, despite changes in firing rates of individual neurons (*Murray et al., 2017a*).

One potential explanation for the differences in coupling between LIP and FEF (*Figure 3*) could be due to differences in receptive field overlap between pairs of neurons. However, the finding that differences in coupling exist even during fixation (*Figure 6a*) suggests true differences in the underlying functional organization of the areas rather than simply an artifact of sampling. To explore the relationship between spatial selectivity and coupling further, we analyzed the coupling between neurons as a function of their spatial tuning similarity based on the distance between the peaks of the their 1D tuning curves (*Figure 6—figure supplement 1*). We found that coupling was modestly

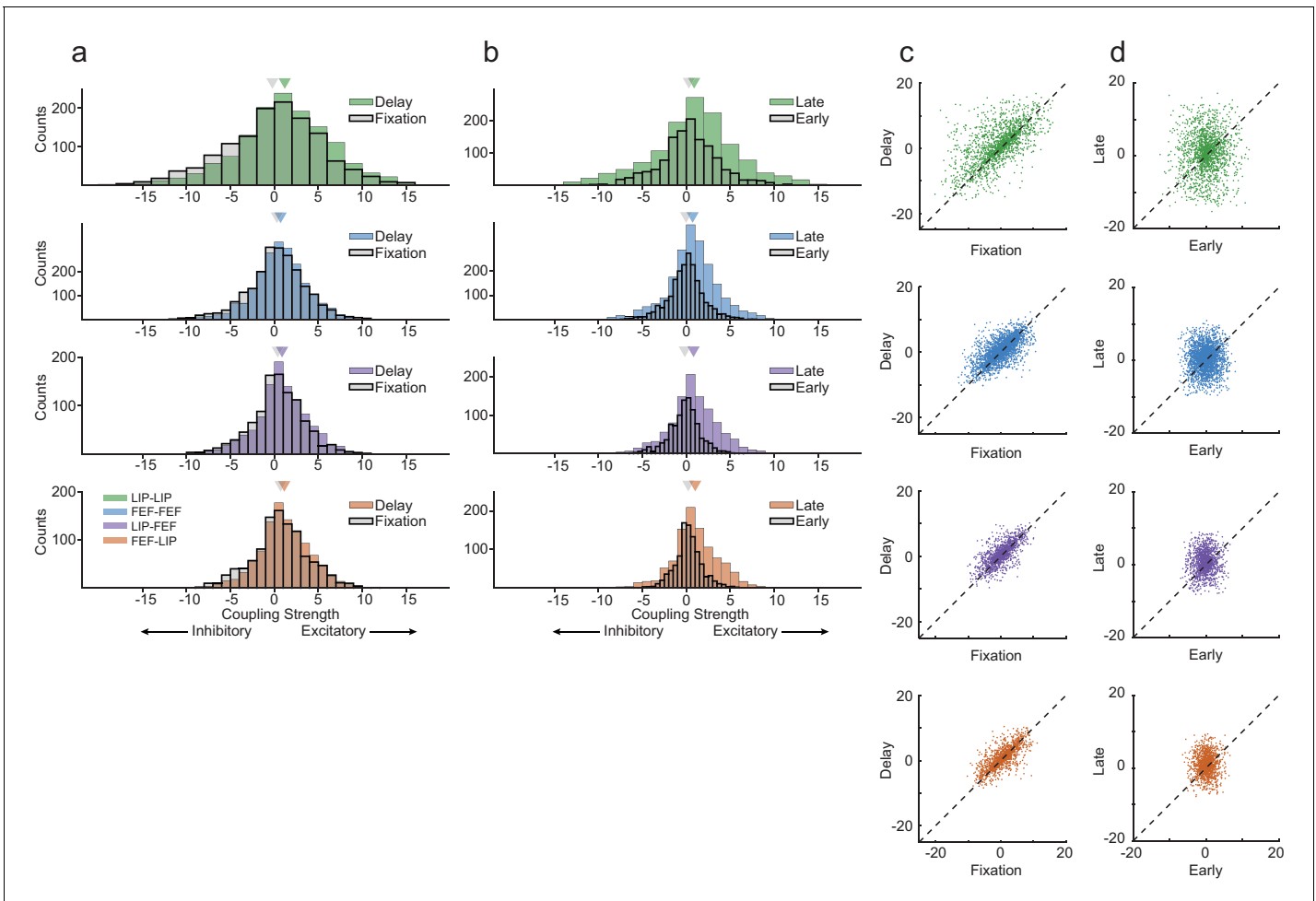

**Figure 6.** Interneuronal coupling is dynamic across behavioral epochs. (**a**) The strength of coupling increased between the fixation period and delay period across all area interactions, especially within LIP (Wilcoxon rank-sum test, p=4.87×10$^{-09}$; p=5.17×10$^{-04}$; p=1.64×10$^{-04}$; p=3.01×10$^{-03}$). Strong coupling was observed even during fixation. (**b**) Coupling increased from early in the delay period to late in the delay across all area interactions (Wilcoxon rank-sum test, p=1.08×10$^{-06}$; p=1.65×10$^{-27}$; p=3.96×10$^{-23}$; p=1.97×10$^{-17}$). LIP exhibited increases in excitatory and inhibitory interneuronal interactions, whereas FEF and between area interactions were dominated by increases in excitatory connectivity. (**c**) Interneuronal coupling between the fixation and the delay period was systematic (Pearson's r, R = 0.61; R = 0.66; R = 0.70; R = 0.72), (**d**) whereas the coupling strength between early and late in the delay showed no relationship (Pearson's r, R = 0.04; R = 0.04; R = 0.06; R = 0.03).

The online version of this article includes the following figure supplement(s) for figure 6:

**Figure supplement 1.** Pairwise coupling as a function of spatial tuning similarity.
**Figure supplement 2.** Spike-history dynamics over behavioral epochs.

stronger for pairs of neurons with more similar spatial tuning, which was most pronounced for neurons within LIP. Interestingly, this feature is predicted by classic bump attractor models of spatial working memory.

To assess the temporal stability of the network interactions across the delay period, we compared coupling early in the delay period to coupling late in the delay period. Coupling increased late in the delay period for all interactions (*Figure 6b*; Wilcoxon rank-sum test, LIP-LIP, p=1.08×10$^{-06}$; FEF-FEF, p=1.65×10$^{-27}$; LIP-FEF, p=3.96×10$^{-23}$; FEF-LIP, p=1.97×10$^{-17}$). LIP exhibited the smallest change overall, but a closer view of the distribution of within-LIP coupling kernels revealed a relatively symmetrical increase in excitatory and inhibitory weights. In contrast, within-FEF and between-area coupling reflected primarily excitatory shifts (*Figure 6b*).

Although the increase in coupling between the fixation period and delay period was subtle, it was systematic across all interactions within and between areas (*Figure 6c*). However, unlike the pair-by-pair relationship between fixation and delay coupling, there was little correlation between coupling early and late in the delay period (*Figure 6d*). Notably, within-LIP activity had the largest increase in excitatory coupling from the fixation period to the delay (*Figure 6a*), but the smallest change from early to late in the delay period (*Figure 6b*). This temporal pattern suggests that the network dynamics within LIP are engaged first and then the larger circuit becomes more strongly coupled as the delay period goes on.

We also examined the dynamics of single neurons' spike-history across the temporal epochs. The strength of the spike-history kernels remained consistent between the fixation and delay periods as one might expect (*Figure 6—figure supplement 2a,b*), but interestingly many neurons in FEF had larger spike-history effects later rather than earlier in the delay period (*Figure 6—figure supplement 2c,d*), meaning that their autocorrelation increased over the delay. Crucially, remember that the impact of the history term is estimated while controlling for the drive of the saccadic response.

## Coupling differences in LIP and FEF can be explained by the strength of recurrent connectivity in attractor networks

To further explore the implications of our results for mechanistic models of persistent activity, we fit the population GLM to spiking data from a classical 'bump' attractor network used to model spatial working memory (*Figure 7a,b*; *Gerstner et al., 2014*; *Compte et al., 2000*). Our main goal was to attempt to infer the recurrent connectivity of the attractor network using the GLM to provide a benchmark to relate the data to the mechanistic models (or perhaps vice versa). We parametrically varied the recurrent connection strength of the excitatory population in the attractor network while holding all other parameters constant (at their default values) to generate synthetic datasets from each of these network variants. We then sampled spike trains from a subset of the total network population, akin to a real experiment (20 neurons, 500 trials per session). To further mimic the real experiment, we varied the onset and location of the stimulus on a per trial basis, and the sample of neurons remained the same throughout each session. We fit the population GLM to the spike trains of the ensemble of neurons in each session in the exact same way we did for the real datasets.

The coupling between neurons (determined by the GLM) systematically increased with greater recurrent connectivity in the attractor network, confirming that coupling can act as a valuable proxy for the strength of the recurrent weights of the population (*Figure 7c*). The black dashed line in *Figure 7c* indicates the level of coupling at the minimum recurrent strength necessary to generate persistent activity. The green and blue lines are LIP and FEF's true coupling determined from the data, respectively. Given these simulations, LIP appears to exhibit strong attractor dynamics (i.e. near the point of saturation of recurrent functional connectivity), whereas FEF acts as a weak attractor, but one still independently capable of maintaining persistent activity (i.e. near the model's theoretical minimum for exhibiting persistent firing, at least given the other parameter settings). The coupling we observed between LIP and FEF was statistically significant, but lay below the threshold to be able to generate persistent activity on their own (*Figure 7—figure supplement 1*). Data from area MT (from other experiments, described earlier) had even lower coupling (*Figure 7—figure supplement 1*), consistent with that area's role in visual function and lack of strong persistent responses.

The stimulus, history, and coupling kernels fit to spike trains from the attractor network all resembled those recovered from the real data. The GLM captured the effects of the stimulus, both when it was in or out of the receptive fields of the sampled neurons (*Figure 7d*, panel 1). The history term recovered the 'built-in' hard refractory period of the neurons, as well as the longer autocorrelation

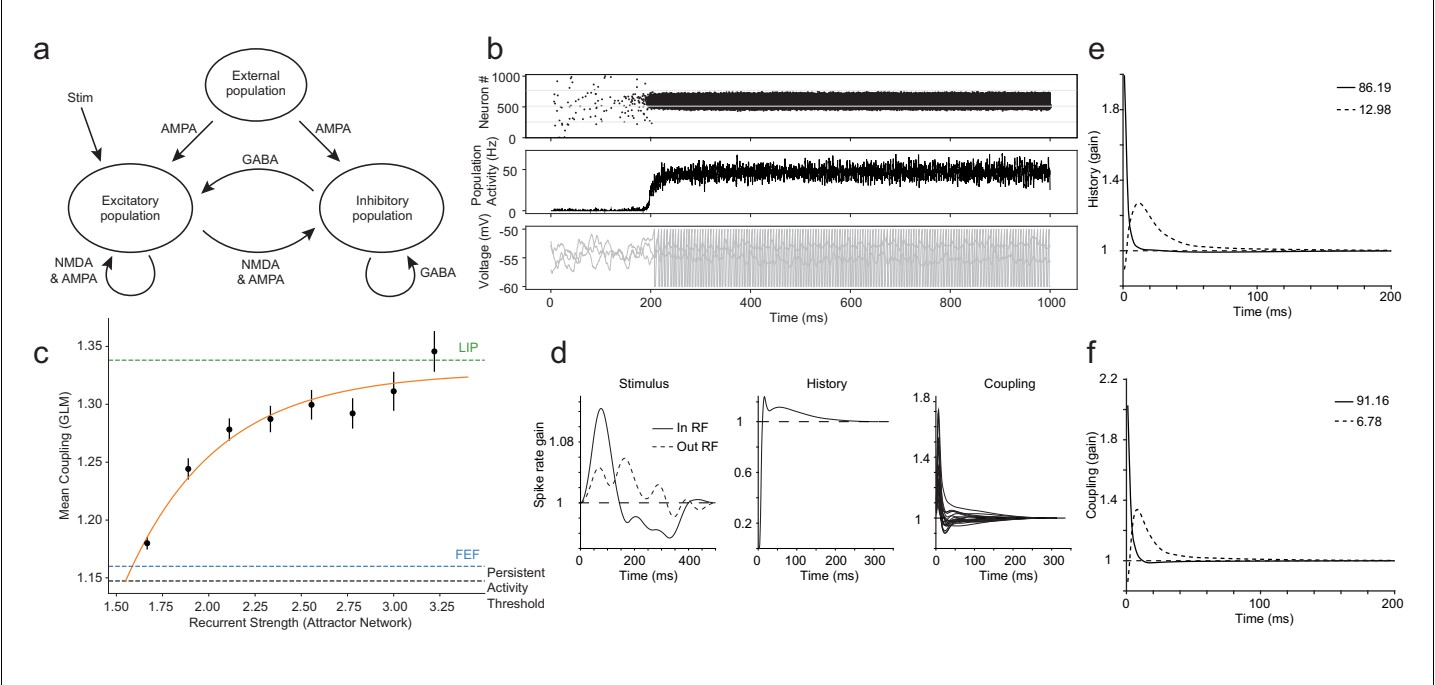

**Figure 7.** Coupling differences in LIP and FEF can be explained by the strength of recurrent connectivity in attractor networks. Synthetic datasets were generated from a classic attractor network and fit with the GLM using the same procedure used with the real data. (**a**) Schematic of classic attractor network used to model spatial working memory (adapted from *Gerstner et al., 2014*). (**b**) Population activity of attractor network for an example session with high recurrent strength (2.3): a stimulus triggers selective persistent activity (top, middle row); voltage traces from three example neurons (bottom row). (**c**) A systematic relationship was revealed between the mean coupling (estimated from the fitted GLM) and the internal recurrent strength of the attractor network (error bars = SEM). The black dashed line indicates the level of coupling at the minimum recurrent strength necessary to generate persistent activity. The green and blue lines are LIP and FEF's coupling determined from the data, respectively. (**d**) Stimulus, spike-history, and coupling kernels of an example neuron. PCA of history (**e**) and coupling (**f**) kernels for an example session revealed the 'fast' and 'slow' dynamics of the neurons specified in the network, which accounted for virtually all the variance explained (legend).

The online version of this article includes the following figure supplement(s) for figure 7:

**Figure supplement 1.** The relationship between coupling (GLM) and recurrent strength (attractor network) for the other area interactions and the shuffling control.

**Figure supplement 2.** All history and coupling kernels from an example session from the attractor network.

(*Figure 7d*, panel 2). The coupling terms captured the strong and prevalent interneuronal correlations (*Figure 7d*, panel 3), although coupling tended to be faster in the attractor network than in the data, which may be an interesting difference worthy of consideration in modeling. The first two principal components (*Figure 7e,f*) of all the history kernels and coupling kernels (*Figure 7—figure supplement 2*) explained virtually all of the variance, and revealed the temporal profiles of the fast and slow excitatory components (AMPA and NMDA receptors, respectively), which are explicitly specified as part of the synaptic weights of the population in the attractor network. It is unsurprising that the coupling and history kernels are similar, given that the artificial neurons in the population are all identical and the profile of connection weights yokes the between-neuron connection strength and self-recurrent strength in the attractor network.

These findings from the GLM analysis of the attractor network serve not only to verify the inferences that came from the core analysis scheme as applied to real data, but also to help situate our results more broadly with respect to theoretical models. The GLM recovered reasonable stimulus, history, and coupling kernels from the artificial population, and captured the underlying strength of recurrent connections in the attractor network. Our initial comparisons also revealed that LIP and FEF both fell within the range of theoretically-predicted couplings required for persistent activity, although at intriguingly different positions within that range. There are, of course, numerous parameter tradeoffs that can be explored in this and other similar network models (*Chaudhuri and Fiete, 2016*; *Zylberberg and Strowbridge, 2017*). The analyses reported above are only the first steps to

interrogating attractor networks in detail and linking the theories back to data, but they demonstrate the exciting feasibility and validity of using a population GLM to link theoretical spiking models with recordings of neuronal ensembles.

## Discussion

The primate frontoparietal network is often put forth as a model system in which to study cognitive processes, yet little is known about the circuit-level spiking activity within and between regions in this network. We therefore characterized the interneuronal interactions in this circuit during oculomotor working memory using multi-area multi-site simultaneous recordings, and a population encoding framework that extended the generalized linear model to simultaneous fits of multiple neurons recorded at the same time in two cortical areas. This enabled us to isolate the contributions of network and intrinsic dynamics to persistent activity while simultaneously disambiguating the effects of task variables.

Historically, models of cortical persistent activity have focused on the challenges posed to generic architectures, identifying particular network interactions and cellular properties whose interplay might be capable of maintaining the sorts of persistent activity seen in the brain (for review: *Wang, 2001*). Such models have emphasized strong recurrent activity (to maintain spiking without additional input), broadly tuned inhibition (useful for damping the runaway dynamics), and a degree of (realistic) slow intrinsic time constants to stabilize the network (perhaps mediated by NMDA receptors). Using a population encoding model, we were able to identify several of these key features in the population activity, and to map these on to data from specific brain areas.

We found evidence of strong reciprocal interactions both between and within FEF and LIP. These circuit-level interactions comprised three main components. First, the slow timescale excitation we observed confirms the existence of strong reverberatory activity, the primary functional component used in models to generate persistent activity. Second, slow inhibitory interactions were present that could serve to perform the hypothesized tempering of reverberatory excitation, curbing potentially runaway attractor dynamics. Finally, we observed a fast excitation component, primarily within areas, which corresponds to more conventional local neuron-to-neuron interactions. The magnitude and time courses of all components serendipitously map on to three key biophysical features that are prevalent in theoretical models of persistent activity: NMDARs, GABARs, and AMPARs (*Compte et al., 2000*; *Goldman-Rakic, 1995*; *Wang, 1999*). Of course, our characterizations provide summaries of temporal scales at the level of spikes, and how these quantitatively relate to intracellular mechanisms invites further study. Recent work suggests a more complicated story at such lower biophysical levels (*Rodermund et al., 2020*). However, our validation of the GLM analysis with the attractor network lends credence to the estimates of the neurons' temporal dynamics. The time constants of neuron-to-neuron interactions, as well as the history dynamics of individual cells, were on order of 100–250 ms, and are therefore consistent with the relatively long dynamics that are called for theoretically to maintain persistent activity over timescales of many 100 s of milliseconds to a few seconds.

Theories of persistent activity have only recently begun to tackle the multi-area circuit dynamics underlying working memory (e.g. *Murray et al., 2017b*). This distributed frontoparietal model attempts to explain working memory and perceptual decision making with two interconnected modules with differences in the recurrent structure of the local networks. The idea is that stronger attractor dynamics in PFC than PPC leads to more robust persistent activity during working memory, and categorically different computations during decision making (i.e. winner-take-all choice in PFC, and slow temporal integration in PPC). Work by *Murray et al. (2017b)* nicely lays out several potential scenarios for the local and long-range network structure that could support attractor dynamics in the frontoparietal circuit. Our GLM analysis of the attractor network provides the unique opportunity to speak directly to these architectures. Both LIP and FEF appear to be independently capable of stable attractor dynamics, whereas the between-area interactions fell below the threshold for independently generating persistent activity, despite significant coupling (this is the first option: row 1 of *Figure 3c* of their paper). However, unexpectedly, LIP had much stronger recurrent connectivity than FEF, which runs counter to the initial proposal for the specific differences in local structure in the two areas. Thus, our results suggest that the proposed underlying differences in network

structure are unlikely responsible for the theoretical functional differences between LIP and FEF in this distributed frontoparietal model (or at least not in the expected way).

Although our results suggest that LIP has a stronger degree of locally-recurrent connectivity, the strong interactions both between and within these areas on single trials support the idea of a tightly linked network which might be better thought of as parts of a single functional unit. The observed timescales of recurrent excitation between neurons likely indicate evidence for attractor-like models of network dynamics, not only within LIP and FEF, but across the entire circuit, which could be effectively acting as a single distributed attractor with only quantitative differences across areal nodes. The superior colliculus and thalamus are also likely critical parts of this circuit (*Ferraina et al., 2002*). Recent work in rodents has suggested that the thalamus plays a central role in maintaining persistent activity (*Bolkan et al., 2017*; *Guo et al., 2017*; *Inagaki et al., 2019*; *Schmitt et al., 2017*), a proposition that needs to be tested in the primate, and which might further enrich our perspective on the full network (or networks) involved in even simple instances of persistent activity.

Taken together, our results provide direct empirical support for strong recurrent excitation and balancing inhibition in primate brain areas that exhibit persistent activity. These findings inform and constrain the structure of the connectivity, both qualitatively and quantitatively. An important next step will be to test a multi-area mechanistic model that incorporates attractor dynamics within and between areas using magnitudes and timescales informed by our data, and which takes into account the differences in architecture and network/intrinsic drive in each area. However, a crucial difference between the data and the model may lie in the fact that bump attractor networks fundamentally link the spatial connectivity of the network to the spatial selectivity profile of the neurons in the population, but relatively little is known about the fine-grained retinotopy (and microcircuit architecture) in area LIP (*Arcaro et al., 2011*). Augmenting the attractor network model by incorporating heterogenous tuning functions and/or extending the model to capture 2D receptive field profiles would be valuable next steps to better understand the nature of the functional architecture in these areas. It is possible that the heterogeneity and mixed selectivity prevalent in this circuit is a result of heavily interconnected networks that are continuously active (*Fusi et al., 2016*; *Meister et al., 2013*; *Rigotti et al., 2013*). These features may be critical for normal oculomotor behavior, in which the eyes move almost constantly, and sensory inputs must be continuously and flexibly mapped to motor actions.

## Materials and methods

### Stimulus presentation apparatus

Macaques were seated, head-fixed, in custom primate chairs facing a rear-projection screen (IRUS; Draper; 150 cm $\times$86 cm) at a distance of 57 cm, which covered the central 106∘$\times$73∘ of visual angle. Images were projected onto the screen by a PROPixx projector (VPixx; resolution = 1920 $\times$ 1080 pixels, refresh rate = 120 Hz). Visual stimuli were generated using Matlab (MathWorks) and Psychtoolbox (*Brainard, 1997*; *Kleiner et al., 2007*). Neural data and task events were recorded with the OmniPlex system (Plexon) using a Datapixx I/O box (VPixx) for precise temporal registration. Electrical microstimulation was administered with a PlexStim device (Plexon). All hardware and software were integrated using the PLDAPS experimental framework (*Eastman and Huk, 2012*). Eye position was tracked using an Eyelink 1000 video eye tracker (SR Research), sampled at 1 kHz. Reward was delivered through a computer-controlled syringe pump (New Era).

### Memory-guided saccade task

Macaques were trained to perform a classic visuospatial delayed-response task: memory-guided saccades. On a trial, the monkey fixates on a central point while a target is flashed (for 200 ms) at a random location in the periphery. After a delay period, in which he must maintain fixation, a 'go signal' is indicated by extinguishing the fixation point, and he must make a saccade to the remembered location of the target. There are two key differences in our specific version of this task: 1) the timing of the task events: the fixation point onset (1–1.5 s durations), target onset, and fixation off (go signal) are variable, making the delay period variable on a trial-by-trial basis (delays = 0.5 s - 2 s), and 2) the spatial location of the target is jittered continuously (eccentricities were drawn from a Gaussian distribution, mean = 10˚, SD = 5˚), rather than being presented at one of an array of fixed grid

locations. The spatial jitter was applied to the position of the target in X and Y in the visual field (in degrees of visual angle), not radially from the center. The delay durations (as well as other task timings) were constructed from uniform distributions. This task served a dual purpose: to functionally map LIP and FEF, and as our primary way to investigate persistent activity during working memory.

## Preparation and electrophysiology

We performed electrophysiological recordings in cortical areas LIP and FEF in two male rhesus macaques (9 y.o., 10.1 kg, and 5 y.o., 6.8 kg). Each monkey had two custom titanium chambers implanted over the regions of interest (ROIs) to allow for simultaneous recordings in both areas. Chambers were placed according to structural MRI scans and anatomical landmarks. An accurate 3D model of the skull (both digitally, and 3D-printed) was used to craft an individualized fit of the chamber to the skull for a more stable and sealed implant. All experimental protocols were approved by The University of Texas Institutional Animal Care and Use Committee and in accordance with National Institute of Health standards for care and use of laboratory animals.

Extracellular electrophysiological recordings were conducted with multisite linear electrode arrays (U-probes; Plexon; 24 or 32 channels) with stereotrode geometry (50 µm between adjacent sites; 100 µm between pairs). Probes were lowered daily into both chambers. ROIs were identified using electrode depths, sulci landmarks (using the physiological characteristics of white/gray matter), neuronal properties, and in the case of FEF, electrical microstimulation. Both LIP and FEF were functionally identified and mapped by observing spatially selective visual and saccadic activity, as well as delay-period persistent activity in the memory-guided saccade task (*Gnadt and Andersen, 1988*). In addition, FEF was mapped with a well-established microstimulation protocol (*Kimmel and Moore, 2007*; *Moore and Fallah, 2001*), in which conventional currents (~50µa, 200 Hz) tended to generate fixed vector saccades to regions of visual space.

## Neural analysis

We collected a wide sample of units in LIP and FEF (967 units; LIP: 407; FEF: 560; Monkey A: 459; Monkey L: 508; 160 sessions; 7448 pairs; mean firing rates: FEF: 36 sp/s; LIP: 28 sp/s; $p=1.16 \times 10^{-5}$ Wilcoxon Rank-Sum test). Offline spike sorting was conducted in a semi-automatic fashion, first with KiloSort (*Pachitariu et al., 2016*), and then manually in Phy. Clusters were merged or split based on the units' characteristics: waveforms, ISI distributions, temporal continuity, and principal components. Peri-stimulus time histograms (PSTHs) were constructed by aligning the spike times to the major task events (target onset or saccade), binning at 1 ms resolution, averaging across trials, and smoothing with a Gaussian filter (SD = 30 ms). MT data were collected during a different experiment under the same behavioral conditions in prior reports (*Yates et al., 2017*).

## Encoding model

We built and compared several population encoding models, which were all variants of a generalized linear model (GLM). This statistical approach allowed us to characterize how a neuron's spike trains (on single trials) depended on the task events (the visual stimulus and motor response), spikes from simultaneously recorded neurons, and the fitted neuron's own recent spike-history. In this way, we could infer the functional connectivity of the network on a trial-by-trial basis. Each daily session (including all simultaneously recorded neurons) went into the GLM analysis. The data (and fits) were then summarized across all sessions.

Spike trains were discretized into 1 ms bins ($\Delta$). Covariates were represented as delta functions at the time of the event and parameterized with temporal filters (nonlinearly time-scaled raised cosine functions) following previous conventions (*Pillow et al., 2008*; *Park et al., 2014*; *Yates et al., 2017*). The conditions inside and outside the receptive field were modeled together along with the rest of the parameters. The model does not include details about the specific saccade location or the stimulus spatial jitter. The number of parameters and data points depends on the number of simultaneously recorded neurons in each session (and the length of the trials which varies; time bins = 1 ms). For each neuron, the target and saccade kernels are each 8 parameters and 800 ms long, and the history and coupling filters are each 10 parameters and 250 ms long. There are *n*-1 coupling filters for each neuron, where *n* is the number of simultaneously recorded neurons during each session.

These parameterizations are conventional and conform to previous uses of this method (*Pillow et al., 2008*; *Park et al., 2014*; *Yates et al., 2017*).

Each neuron was fit with a GLM using maximum a posteriori estimation (MAP) with a log-Gaussian prior, where the instantaneous spike rate (conditional intensity) of the fully coupled model at time $t$ is given by

$$\lambda_t = \exp(\mathbf{k} * \mathbf{x}_t + \mathbf{h} * \mathbf{r}_{t-1} + \mathbf{c} * \mathbf{s}_t + \mathbf{b})$$

where $\mathbf{k}$ comprises the $n_i$ weights on the $m$ task covariates $\mathbf{x}$, and $\mathbf{k}*\mathbf{x}$ is shorthand for

$$\sum_{i=1}^{m}\sum_{j=1}^{n_i}\mathbf{k}_{ij}f_j(\mathbf{x}_i(t-\tau:t))$$

where $f$ is the temporal basis functions described previously. $\mathbf{h}$ is the weights on the recent spike-history (which is notation for the neuron's spikes leading up to but excluding the current time bin), $\mathbf{c}$ is the coupling weights on $\mathbf{s}$ simultaneously recorded spikes, and $\mathbf{b}$ is a constant term that captures the neuron's baseline firing rate. We used an exponential as our choice of nonlinearity (inverse-link function) to map the linear predictors to Poisson spiking.

The log-likelihood of a Poisson GLM for a single neuron is

$$L(\lambda;\mathbf{r}) = \sum_t \mathbf{r}_t log(\Delta\lambda_t) - \Delta\lambda_t$$

Only simultaneously recorded neurons within each session were included in the full (coupled) population encoding models. To prevent overfitting, we used ridge regression to penalize the weights on the regressors, which has the effect of shrinking the coefficients.

$$\alpha = (\|\mathbf{k}\|_2 + \|\mathbf{h}\|_2 + \|\mathbf{c}\|_2)$$

For each neuron, we performed 5-fold cross-validation, in which the model was trained on 80% of the data and tested on the withheld 20%. The goodness-of-fit was computed by generating spike trains from the model, averaging them to get a predicted PSTH and calculating the variance explained of the real PSTH (R-squared).

To evaluate and compare the performance of the models, we calculated a likelihood test statistic by taking the difference of the log-likelihood for the full model and the log-likelihood of a homogeneous Poisson process model (with a constant rate), which acts as a null or 'saturated' model (*Runyan et al., 2017*),

$$LL = (L(\lambda;\mathbf{r}) - L(\bar{\lambda};\mathbf{r}))/\sum_t \mathbf{r}_t$$

where $\bar{\lambda}$ is the mean firing rate of the neuron (over the entire trial) and we divide by the sum of the spike counts. This formulation gives the prediction accuracy in units of bits per spike (*Paninski et al., 2004*; *Pillow et al., 2008*).

The models we considered varied in their inclusion of the coupling, history, and stimulus terms, as well as their parameterizations. We verified the fits by comparing the variance explained of the PSTHs and single-trial spike trains between the full model (with coupling, history, and task terms) and an 'uncoupled' model, which had no coupling terms (i.e. no connections between the neurons) but was otherwise the same. Unsurprisingly, adding coupling to the model (which captures the correlational structure of the single-trial spike trains) does not improve the fit of the predicted PSTHs (*Figure 3d*; points fell around the unity line; median 76% variance explained under both models) because PSTHs (by their very nature) ignore single-trial dynamics by averaging across trials. However, crucially, predictions of the single-trial spike trains universally benefit from the inclusion of the coupling terms (*Figure 3e*; points fell above the unity line; on average a log-likelihood improvement of 22% in units of bits per spike).

We also compared models that only included spikes from different temporal epochs of the trial to identify neural population dynamics that varied as a function of task demands. In the cases where two temporal epochs were compared, the window sizes and task parameters were matched. To summarize the coupling results within and between ROIs, we performed principal component analysis

(PCA) on the coupling kernels sorted by each interaction. The first three principal components explained around 90% of the variance for each interaction. The same was true for the history kernels. To estimate the history and coupling time constants, we fit a double exponential curve to each kernel or the first principal component of the kernels for each ROI. History kernels exhibiting strong refractory periods rather than long timescale dynamics were poorly fit by an exponential decay and were excluded from this analysis (48%). To further characterize the impact of the coupling terms for each neuron, we constructed a predictive index defined by the difference between the deviance explained by the full model, and the deviance explained by the model without coupling (the 'uncoupled' model— only task and history predictors) divided by the deviance explained by the coupled model (*Runyan et al., 2017*; *Agresti, 2002*). A coupling index of one indicates that the neuron's spike trains are completely predicted by the spiking activity of other neurons, and a value of zero indicates that spikes from other neurons add no predictive power over and above the task predictors and the neuron's recent spike-history. We constructed the analogous index for the spike-history as well.

To further validate the coupling, we constructed null distributions by fitting the GLM to permuted data (following *Pillow et al., 2008*; see supplement). We permuted the order of the trials (i.e. trial identity) for each neuron (whose spike trains are used as predictors) relative to each fitted neuron, which eliminates the correlations between neurons but maintains the temporal dependencies of the task events and the spike times *within* each trial for the fitted neuron. In other words, the signal correlations are preserved while the noise correlations are disrupted. Crucially, the task events and spike times are never misaligned on single trials (on which the model is fit). This procedure is equivalent to treating each neuron independently in the model and obscuring the simultaneous nature of the recordings. The permutation analysis provides a clear null model against which to identify functional interactions within the GLM (i.e. beyond what is expected by chance) and is a principled validation of the coupling (*Pillow et al., 2008*; *Yates et al., 2017*; *Runyan et al., 2017*).

An additional concern might be that the differences in coupling between areas is driven by differences in receptive field overlap due to artifacts of sampling. The overall differences in coupling between LIP and FEF were also present during fixation when there is no visual stimulus in the receptive fields or memory of the stimulus being maintained (*Figure 6a*), meaning the coupling differences are likely due to differences in the underlying functional connectivity and not due to sampling artifacts regarding the receptive fields (which concurs with previous work: *Katsuki et al., 2014*). In addition, experiments were conducted using identical methods in both areas during simultaneous recordings and using a relatively indiscriminate sampling protocol (all neurons with isolatable waveforms were included regardless of selectivity). We also quantified the effect of coupling as a function of tuning similarity (using 1D circular Gaussian tuning curves), and found that coupling was modestly stronger for pairs of neurons with more similar spatial tuning across all interactions, and that relationship was most pronounced for neurons within LIP (*Figure 6—figure supplement 1*), a feature predicted by classic attractor network models. However, to fully address this question, the extent, shape, and response profile of the response fields would need to be mapped in greater spatial detail.

In addition, differences in coupling between the ROIs were unlikely due to overall differences in mean spike rates. The coupling kernels for each area were compared to the null (shuffled) distributions within each respective area. Under both conditions, the spike rates were identical but during the shuffled condition the correlation structure between neurons was broken, providing an independent estimate of coupling due to chance alone for each area regardless of mean spike rates. We have also demonstrated this in the GLM analysis of the simulated data from the attractor network (*Figure 7—figure supplement 1*), in which coupling tracked the internal recurrent strength of the attractor network independent of overall changes in mean firing rate. These results are further bolstered by the data– neurons in FEF actually had a significantly higher mean firing rate than LIP, whereas LIP had much stronger coupling than FEF.

Area MT's coupling was estimated using the same GLM analysis, but the data was collected during a perceptual decision making task rather than a working memory task (details described in *Yates et al., 2017*). Coupling within MT was significantly lower than each other area interaction (*Figure 3—figure supplement 2*): LIP-LIP, FEF-FEF, LIP-FEF, FEF-LIP, (Wilcoxon rank-sum test): p=$2.19\times10^{-143}$; p=$6.92\times10^{-103}$; p=$2.18\times10^{-07}$; p=$7.98\times10^{-22}$. Magnitude of the effect size

(Cohen's D): LIP-LIP = 0.89 (large effect), FEF-FEF = 0.41 (medium effect), LIP-FEF = 0.16 (small effect), FEF-LIP = 0.24 (small effect).

## Attractor network

To validate our analyses and situate our results more broadly we generated synthetic datasets from a spiking attractor network (*Gerstner et al., 2014*; https://neuronaldynamics.epfl.ch/) and fit them with the GLM. This attractor network was based on a classic model of spatial working memory (*Compte et al., 2000*). We parametrically varied the strength of the recurrent connections in the excitatory population while holding all other parameters constant. To replicate the real experiments, we varied the location and onset of the stimulus (i.e. the 'visual target') trial-by-trial (200 ms duration; onset jittered from 200 to 500 ms; 1 s trial duration), and sampled a small subset of neurons from the overall population (N = 20). 500 trials were simulated per session and this sample of neurons remained the same throughout the session. Parameters of the network were kept at their default values unless otherwise noted, as in the case of the recurrent strength of the excitatory population. Some of the main parameters of interest are noted (using their names from the simulation code): N_excitatory = 1024, N_inhibitory = 256, N_extern_poisson = 1000, poisson_firing_rate = 1.3 * hertz, weight_scaling_factor = 2.0, sigma_weight_profile = 20.0, Jpos_excit2excit = 1.6 (strength of the recurrent input), stimulus_center_deg = 120, stimulus_width_deg = 30, stimulus_strength = 0.06 * b2.namp. Note that the default NMDA time constant was 65 ms (*Gerstner et al., 2014*) rather than the 100 ms in the original model (*Compte et al., 2000*).

## Acknowledgements

We thank A Levi for sharing the MT data, C Badillo for technical support, and A Laudano, K Mitchell, and C Carter for animal support. We also thank E Salinas, A Compte, and an anonymous reviewer for their helpful comments, and W Gerstner, W M Kistler, R Naud, and L Paninksi for making the attractor network code available. This research was supported by a National Science Foundation grant (IIS-173491) to ACH and Il Memming Park (Stony Brook University) and a National Eye Institute grant (R01-EY017366) to ACH, and a National Eye Institute training grant (T32-EY021462).

## Additional information

### Funding

| Funder | Grant reference number | Author |
|---|---|---|
| National Science Foundation | IIS-173491 | Alexander C Huk |
| National Eye Institute | R01-EY017366 | Alexander C Huk |
| National Eye Institute | T32-EY021462 | Eric Hart |

The funders had no role in study design, data collection and interpretation, or the decision to submit the work for publication.

### Author contributions

Eric Hart, Conceptualization, Data curation, Software, Formal analysis, Validation, Investigation, Visualization, Methodology; Alexander C Huk, Conceptualization, Supervision, Funding acquisition, Project administration

### Author ORCIDs

Eric Hart (ID) https://orcid.org/0000-0002-6042-8389
Alexander C Huk (ID) https://orcid.org/0000-0003-1430-6935

### Ethics

Animal experimentation: All experimental protocols were approved by The University of Texas Institutional Animal Care and Use Committee (AUP-20418-00063, AUP-2015-00068) and in accordance with National Institute of Health standards for care and use of laboratory animals.

### Decision letter and Author response
Decision letter https://doi.org/10.7554/eLife.52460.sa1
Author response https://doi.org/10.7554/eLife.52460.sa2

---

## Additional files

### Supplementary files
• Transparent reporting form

### Data availability
Data files have been deposited to Data Dryad under https://doi.org/10.5061/dryad.gb5mkkwk7.

The following dataset was generated:

| Author(s) | Year | Dataset title | Dataset URL | Database and Identifier |
|---|---|---|---|---|
| Hart E, Huk AC | 2019 | Data from: Recurrent circuit dynamics underlie persistent activity in the macaque frontoparietal network | https://doi.org/10.5061/dryad.gb5mkkwk7 | Data Dryad, 10.5061/dryad.gb5mkkwk7 |

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
