## [Decision Letter]

**Acceptance summary:**

This study combines multi-neuron recordings from two cortical areas, FEF and LIP, with advanced fitting techniques (GLM modeling) to investigate the underlying neuronal dynamics that unfold during a classic memory-based visuomotor task. The approach is notable because it teases apart the contributions of task events, firing history, and functional coupling across neurons to the evoked spiking activity, doing so in a way that is interpretable. The observed neuronal interactions (magnitudes and timescales) are broadly consistent with those proposed by classic theoretical models of persistent activity and working memory, but the data also reveal intriguing differences and similarities between FEF and LIP that should serve to constrain and refine future models of the frontoparietal attentional network.

**Decision letter after peer review:**

Thank you for submitting your article "Recurrent circuit dynamics underlie persistent activity in the macaque frontoparietal network" for consideration by *eLife*. Your article has been reviewed by three peer reviewers, including Emilio Salinas as the Reviewing Editor and Reviewer #1, and the evaluation has been overseen by Timothy Behrens as the Senior Editor. The following individual involved in review of your submission has agreed to reveal their identity: Albert Compte (Reviewer #2).

The reviewers have discussed the reviews with one another and the Reviewing Editor has drafted this decision to help you prepare a revised submission.

Summary:

This manuscript analyzes single-neuron activity in the FEF and area LIP during the classic memory-guided saccade task. The novelty here lies in the recording of multiple neurons (5-20 per session) from both areas simultaneously, and in the GLM analyses, which are able to tease apart how task events, firing history, and interneuronal coupling determine the spikes fired by each neuron. The results provide a statistical view of the recurrent interactions between and within these two areas that is of relevance to their functional characterization, and to models of recurrent circuits and working memory in general.

All reviewers agreed that this is fantastic data set and that the analysis is interesting and novel. The general concern was that the methods require more thorough validation in order to put the interpretations on a more solid footing. The suggested revisions will require additional analyses, but not additional experiments. The consensus was that they are important for supporting the conclusions (which could change as a result).

Essential revisions:

1) All reviewers identified potential problems with the analysis/interpretation of the results of the GLM procedure, as can be seen in the individual reviews below. However, as detailed by reviewer 2, many of these concerns could be addressed by testing the GLM methods against synthetic data from the very network models that the results seem to support. This would require substantial work, but is in principle straightforward. This sort of analysis, where the model is tested with designed data sets for which the correct answers are known, would substantially strengthen the credibility of the results and their interpretation.

Some important methodological concerns that could be at least partially addressed in this way are the following:

a) It is not entirely clear why the "exceptionally strong coupling" between neurons contributes in such a subtle way to the predictions of neural activity (i.e., they are not needed to accurately replicate the PSTHs).

b) Verifying the shuffling procedure (null hypothesis) for determining the significance of the coupling terms.

c) It is unclear whether the history-dependent kernels only capture intrinsic neuronal properties, as opposed to influences from other neurons that could be equivalent to history effects.

d) How do the results depend on receptive field overlap? In particular, it is possible that the differences between LIP and FEF are the consequence of different overlap patterns in the two areas.

2) The total coupling strength, which is the net result of adding the positive and negative (inhibitory) kernel weights, is used as a summary measure of coupling between neurons. As elaborated by reviewer 1, this is a potentially misleading quantity, as it does not consider the magnitudes of the excitatory and inhibitory components. The analyses should be run also with alternative measures that take this distinction into account.

3) Consideration of differences across neuronal classes; in particular, whether the results are distinctly different for neurons with significant delay period activity, as mentioned by reviewer 2.

4) There is insufficient validation of the model against traditional analysis techniques, such as cross-correlation analyses.

5) There is insufficient description of some of the analytical procedures (see specific comments below).

Reviewer #1:

This work addresses important problems in oculomotor physiology, is sound, clearly described, and contains many interesting nuggets about the FEF-LIP circuitry. Just to mention one of them, the three interaction kernels provide convincing measures of the time constants of these circuits, confirming long-help ideas based on theory and modeling. Other results, such as the stronger interactions across than within areas, are interesting precisely because they do go a bit against the grain.

I only have one substantial comment. Several analyses are based on the total coupling strength, which is the net result of adding the positive and negative (inhibitory) kernel weights. This leads to the conclusion that "coupling in LIP was greater than FEF, suggesting a greater degree of recurrent connectivity in LIP". But the total strength is a crude measure. If the subtraction of excitation and inhibition gives 1 = 20 – 19 net excitation in one area and 2 = 5 – 3 in the other, is it really fair to say that the coupling is stronger in the latter? Clearly not. Stronger nonlinearities are expected for highly balanced circuits in which both excitation and inhibition are stronger. This intuition is missing in the Discussion paragraphs four and five. In that case, identical excitation in both areas but stronger inhibition in FEF than LIP would be perfectly consistent with stronger winner-take-all dynamics in FEF and more tolerance to spatial ambiguity in LIP, in turn consistent with saccade selection in the former and divided attention in the latter.

My suggestion is to quantify the excitatory and inhibitory interactions separately. For instance, by computing the total excitatory coupling on one hand, and the total inhibitory coupling on the other. If both terms *and* the net result are stronger in LIP than FEF, then that would credibly verify the current conclusion ("greater coupling in LIP than FEF"). But I suspect (from Figure 3B) that the situation is a bit more nuanced. For instance, stronger inhibitory coupling in FEF would be consistent with stronger motor competition and/or narrower visual tuning. Likewise, one could argue that the relevant indicator of stronger or more reliable delay-period activity is not the net coupling (excitation minus inhibition) but the total absolute coupling (i.e., the total summed magnitude of the coupling weights). I think that, in general, consideration of these alternatives beyond the net coupling would provide a broader and more substantial view of the data, and a more nuanced basis for their functional interpretation, without any alterations to the GLM model.

Reviewer #2:

This manuscript provides evidence that recurrent mechanisms in LIP, FEF and between these two areas are engaged during persistent activity in a delayed response task. This is supported by fitting neuronal spiking data simultaneously recorded in these areas with a population encoding model. A GLM fitting procedure estimates parameters for visual and motor dependencies, neuronal history dependence, and coupling kernels between simultaneously recorded neurons. The coupling kernels are reported strong in comparison to history kernels, and contain time scales consistent with excitatory and inhibitory mechanisms typically included in attractor models of working memory.

The manuscript is a valuable and significant contribution to the research on the mechanisms of working memory. For this classic task, a population-level analysis of simultaneous recordings in two cortical areas has not yet been presented, and provides important insights into the relative importance of different mechanisms that have been proposed to support persistent activity: intrinsic mechanisms, local circuit recurrence, and recurrent interactions between areas. The methods applied are aimed at testing this with advanced statistical methods.

My major concerns regarding the conclusions drawn from these analyses are:

1) There is an inconsistency in the manuscript in that, on the one hand, interneuronal couplings do not improve PSTH predictions (Figure 3D) and yet various analyses (Figure 3A, Figure 4C) are interpreted as demonstrating "exceptionally strong coupling", "exceptionally strong recurrent interactions" or that "network dynamics play a more vital role than intrinsic dynamics". If recurrent interactions are removed from an attractor model persistent activity collapses, and PSTHs are profoundly affected. At face value, the findings thus do not seem to represent a direct support of these models. The fact that exceptionally strong couplings do not have an impact in the PSTH should be better explained.

2) I am concerned about the "null hypothesis" in Figure 3A: if trials are shuffled for each neuron before the fit, then neural activity is modeled with spiking data of other neurons obtained in trials with different delay duration, so not aligned to the stimulus and saccade predictors used in the model. Since these events are typically associated with an increase in firing rate, interneuronal couplings will be reduced in this model because of these misalignments. The reading of "strong recurrence" that now emanates from the comparison with this "null hypothesis" might instead result from this misalignment. Maybe a more direct assessment of how strong is the contribution of couplings to the network would be to evaluate what fraction they represent in the sum of all predictor terms in the model.

3) The manuscript emphasizes the role of recurrent interactions in generating persistent activity. However, not all neurons recorded in LIP and FEF show persistent activity in this task. In particular, Figure 2 shows that neurons with robust persistent activity show sustained history kernels, which supposedly affect just a minority of neurons in the database (Figure 4C). Some of the conclusions may change if one considers separately neurons with strong persistent activity from other neurons. For instance, if neurons with persistent activity mostly populate the right cloud of dots in Figure 4C the interpretation of this panel's results would be very different. Example neurons shown in Figure 1 could be explicitly identified in Figure 4C and other panels with individual neuron data. If the question is how persistent activity is generated, then it should be relevant to consider separately neurons with and without persistent activity.

4) The interpretations in the manuscript jump very easily between the statistical description contained in the fitted kernels and biophysical mechanisms, and this should be more carefully considered. Any "temporal integration above and beyond that explained by the external task elements" (Park et al., 2014) and the spike times of the few simultaneously recorded neurons in that session will be attributed to history kernels, which would therefore be inappropriate to attribute to intrinsic cellular mechanisms. In general, authors should be cautious with the possible confusion in using "intrinsic" to refer to the history kernel. The history kernel might include traces of other network dynamics not captured by the few neurons recorded simultaneously in the session.

Importantly, many of these concerns could be directly addressed by analyzing spike trains from a few neurons selected from attractor network simulations (many of which freely available) to verify if their GLM approach is able to extract the biophysically-based parameters of the model, as currently argued. My intuition is that model neurons without any specific intrinsic mechanism with long time course would show a long-lasting history dependence (self excitation) when modeled against the activity of just a handful of their close neighbors. The number of neurons used as predictors, or the similarity of their memory fields, could also conceivably have an impact in the decay time of history and coupling kernels, and thus contribute to possible differences between FEF and LIP in this analysis. These computational models are freely available (for instance https://neuronaldynamics-exercises.readthedocs.io/en/latest/) and the model data can be generated easily and analyzed with the authors' methods.

Reviewer #3:

The paper uses a novel analytical technique to infer functional connectivity among FEF and LIP cells during a memory guided saccade task. The analysis takes a GLM-based approach and fits spike trains of simultaneously recorded FEF and LIP cells as weighted combinations of kernels related to (1) target location, (2) saccade direction, (3) coupling filters modeling interactions among cells and (4) history filter modeling within-neuron autocorrelations. The main conclusions are that FEF and LIP neurons show strong coupling both within and across areas, and that LIP shows stronger coupling and longer intrinsic timescales than FEF.

While the approach is interesting and potentially useful, I am afraid it is not sufficiently justified in the present version of the manuscript. The level of detail provided is simply not sufficient to support the claims based on this novel analysis technique. I see two main problems with the manuscript as it is now:

1) There is insufficient control for potentially differential sampling of neurons in FEF and LIP: Could the apparent stronger coupling in LIP vs. FEF be due to different neural sampling in the two areas? For instance, if the RF of the recorded cells had more overlap in LIP relative to FEF, that may produce stronger apparent coupling in two different ways. Stronger coupling may arise if the model mis-attributes shared stimulus/saccade related activity to the coupling weights. The authors emphasize that their model identified coupling above and beyond any stimulus-related response, but they provide little evidence to substantiate this assertion. Stronger coupling may also arise if the model correctly attributes variance but the neurons are in fact more coupled by virtue of being more functionally similar. In that case, the areal differences are only artefacts of the neurons the authors happened to sample. We need much more thorough analyses of the similarity in RF profiles and temporal response profiles of the recorded cells to rule out a sampling confound.

2) There is insufficient validation of the model against traditional analysis techniques. The field has several traditional, established measures of neural coupling or connectivity, including noise correlations and joint PSTHs or autocorrelograms. The authors dismiss those analyses off hand with little justification except for saying that they do not adequately remove stimulus-evoked activity. But there are actually many ways for them to get around confounds with these methods. Comparing their model results to these traditional analyses is essential to validate their results.

3) There is insufficient description of the analytical procedures:

– How were the neurons combined across recording days? Clearly, the neurons that were simultaneously recorded each day could be fit using the described method. But the authors say that they simultaneously fit the entire data set – including neurons recorded on different days. How precisely did they set this up and how did this affect the results?

– What were the kernels that the authors used in the model fits? How were those kernels selected? What properties must the kernels have to make the model work? Are the conclusions robust over reasonable changes in kernel shape?

– Goodness of fit and model comparisons: The authors should provide a complete description of the type of models they compared.

– Individual monkey results: How many cells were recorded in each monkey? Did the critical results hold in both monkeys?

[Editors' note: further revisions were suggested prior to acceptance, as described below.]

Thank you for re-submitting your article "Recurrent circuit dynamics underlie persistent activity in the macaque frontoparietal network" for consideration by *eLife*. Your article has been reviewed by two peer reviewers, including Emilio Salinas as the Reviewing Editor and Reviewer #1, and the evaluation has been overseen by Timothy Behrens as the Senior Editor.

The reviewers have discussed the reviews with one another and the Reviewing Editor has drafted this decision to help you prepare a revised submission.

Summary:

The authors have been very responsive to the comments made in the original reviews, and have thoroughly revised the manuscript. The methods are clearer and the conclusions and rationale for the model more robust. Importantly, the GLM approach has now been validated by applying it to simulated data, and the new results are largely consistent with the original intuition provided by the authors.

However, several implementation details that can potentially affect the results were still unclear and require more thorough explanation.

Essential revisions:

1) What exactly is the uncoupled model?

2) How exactly is the shuffling across trials performed?

3) More careful analyses of the activity during fixation and early delay.

4) Clarification of the dependence on overall firing rate (separate from the variations in connectivity).

These points are discussed at length by reviewer 2 below.

Reviewer #2:

The revised manuscript contains significant improvements over the previous version, notably the new computational model data that provides support to the methods applied. However, I still maintain some relevant concerns on the controls required to support the interpretations of their analyses as couplings between neurons. This concerns primarily the shuffle approach and the control of firing rate confounds as detailed below.

1) Related to previous Major concern 1.

I realize that part of the confusion here is that the “uncoupled model” is never fully described. I understood that this was the full fitted model, to which couplings were set to zero. From this model I would expect PSTHs to differ largely from measured PSTHs, if couplings are contributing significantly to the GLM. Instead, I gather from Pillow et al. (2008) that the uncoupled model is a reduced model refitted to the data, and then I understand that the fit may be good even if couplings for the full model are strong contributors to the fit. I urge the authors to explain clearly what their uncoupled model is. In fact, in the current revision this is further confused in the Materials and methods (“Unsurprisingly,…”), where the “uncoupled method” is commented in the middle of the description of the shuffle approach. These two separate controls should be clearly distinguished and specified so the reader understands their separate implications. Now this is very confusing.

2) Related to previous Major concern 2.

I am still not convinced about the shuffle approach on which the "extraordinarily strong coupling" result is based. If I understand it correctly, the structure of the data is as schematically depicted in Decision letter image 1, with simultaneous spike trains recorded (here 10 neurons in 10 lines per panel) for separate trials (here 8 trials in 8 panels), where in each trial the stimulus is presented in different locations (here different colors of the simulus and response shadings) and delay durations are also different from trial to trial. Neurons respond with spiking activity depending on whether the stimulus falls within their receptive field or not.

**Decision letter image 1. respfig1:** 

The GLM for the neuron on the gray background would then model spikes based on its specific stimulus and response events for each different trial, on its previous spikes in the corresponding trial, and on the previous spikes of all other neurons in the corresponding trials. Crucially the spikes for these other same-trial neurons are collected based on the same stimulus location, response and delay duration.If I understand correctly the shuffle condition applied in Figure 3, the GLM now models the gray-background neuron based on its specific stimulus and response events for each different trial and on its previous spikes in the corresponding trial, exactly as before, but now the spikes of other neurons are taken from randomly picked trials. This is represented in the schematic in Decision letter image 2 (notice the first lines of each panel are the same for both schematics). If this is the case, then now the spikes for these other shuffled neurons were *not* collected based on the same stimulus location, response and delay duration as the modeled neuron. As a result, the consistency of stimulus activation of neuron 9 and neuron 7 shown in Decision letter image 1 (they are both active or inactive in response to stimuli in all trials), and the consistency of the timing of response events, are broken with the shuffle in Decision letter image 2. This will lead to a radical change in the cross-correlations between these two neurons, even if these spike trains were totally independently generated.

**Decision letter image 2. respfig2:** 

I have run this simulation. Data for these spike trains were independently generated for each neuron (so with no coupling between them), only keeping the consistency of stimulus, response and delay within trials. As shown in the histograms in Decision letter image 3, the mean cross-correlations (as a proxy of the couplings obtained with the GLM) were radically affected by the shuffling procedure. This does not demonstrate any functional coupling between the neurons, but the fact that task events that elicited neural responses were misaligned by the shuffling procedure.

**Decision letter image 3. respfig3:** 

In my view, this effect could explain the extraordinary changes in shuffle histograms in Figure 3. To clarify this, the shuffle should be performed only between trials with equal stimulus location and delay duration, or else apply other commonly used resampling procedures to estimate cross- correlations (Fujisawa et al., 2008; Katsuki et al., 2014), such as within-trial jitter correction methods (Amarasingham et al., J Neurophysiol 2012).3) The section on “Interneuronal coupling is dynamic across behavioral epochs” is very confusing and is not really addressing the point that they are formulating. The fact that there are not very marked changes between couplings measured in the fixation and delay periods is taken as evidence of dynamical working memory without much justification. Later on the comparison of early and late delay periods yields stronger couplings as delay progresses. In my view, this attempt to contribute to the debate on dynamic delay activity, or even activity-silent working memory (Stokes et al., 2015) is overambitious and needs more specific analyses (consider for instance possible traces of the stimuli in the couplings after the response period, by comparing in and out conditions in the baseline period where there is no confound of firing rate for in/out conditions, e.g. Spaak et al., SfN 2017 #339.14; Barbosa et al., Biorxiv, 2019, doi:10.1101/763938). Further, in the interpretations of this paragraph it is disclosed that their measure of coupling is confounded by changes in firing rate (“This finding highlights the potential role of nuanced changes in the dynamics of the network rather than an overall increase in population activity”). This should be declared early on, as this confound may affect many of the interpretations. Any interpretations of difference in coupling strengths should be cautious in previously demonstrating no difference in firing rates, or else in applying more stringent rate-correcting methods such as jitter correction (Fujisawa et al., 2008). Currently, the manuscript does not parallel each of the coupling comparisons (FEF-LIP, fixation-delay, etc.) with corresponding comparisons of mean firing rates, or rate-correcting measures, to support these conclusions.

4) The computational model is a nice demonstration of their methods, in particular of the retrieval of the correct time scale of the interactions between neurons (the AMPA and NMDA current time courses). However, in terms of the strengths of the couplings, the model may also be confounded by changes in firing rate. It will be useful to plot the mean firing rate of model neurons as recurrent connectivity is increased. If this mean firing rate is increasing too, then some compensatory change may be introduced to maintain rates constant as recurrent connectivity increases in order to resolve this concern. The stimulus kernels derived from the model in Figure 7D, left panel, do not seem to correspond to the simulation results: stimuli in receptive fields should generate increase in rates, and stimuli outside the receptive field a suppression of rates.

5) Because of the possible rate confound that affects the coupling measure, the manuscript should provide a mean firing rate description of the various areas and task periods, for instance through average PSTHs.

---

## [Author Response]

Summary:This manuscript analyzes single-neuron activity in the FEF and area LIP during the classic memory-guided saccade task. The novelty here lies in the recording of multiple neurons (5-20 per session) from both areas simultaneously, and in the GLM analyses, which are able to tease apart how task events, firing history, and interneuronal coupling determine the spikes fired by each neuron. The results provide a statistical view of the recurrent interactions between and within these two areas that is of relevance to their functional characterization, and to models of recurrent circuits and working memory in general.All reviewers agreed that this is fantastic data set and that the analysis is interesting and novel. The general concern was that the methods require more thorough validation in order to put the interpretations on a more solid footing. The suggested revisions will require additional analyses, but not additional experiments. The consensus was that they are important for supporting the conclusions (which could change as a result).

We thank the reviewers and editor for the thoughtful feedback, and were very happy that there is shared enthusiasm about the value of these experiments. We are grateful for the chance to both add and clarify validation assessments. These are very important both for making inferences about the data, and for linking those inferences to theories of persistent activity. We describe specifics below, but at a high level, we have addressed this issue in a number of ways.

First, we have clarified the value of the shuffling analyses that we used throughout the first submitted version. There was some misunderstanding on how this validation procedure works, which we have elaborated on in the specific responses, the methods, and in the manuscript. In short, we shuffle the order (i.e., trial numbers) of the trials for each neuron relative to other neurons – which eliminates the correlations between neurons but maintains the temporal dependencies of the task events and the spike times within each trial. This is equivalent to treating each neuron independently in the model and ignoring the simultaneous nature of the recordings. Unsurprisingly, adding coupling to the model (which captures the correlational structure of the single-trial spike trains) does not improve the fit of the predicted PSTHs because PSTHs (by their very nature) ignore single trial dynamics by averaging across trials (a point that we discuss in more detail in specific responses). The shuffling analysis provides a clear null model against which to identify functional interactions within the GLM (i.e., beyond what is expected by chance) and was used as a validation in the original coupled-GLM work (Pillow et al., 2008). It is therefore well-suited to provide the very validation of the coupling required by the reviewers (and ourselves).

Second, we have run the same GLM-based analyses on simulated datasets generated from a conventional attractor network (Figure 7). This is the closest we could come to the spirit of the request to do an analysis where the “correct answers are known” (beyond the null model). There is a wrinkle to this, which is that, in fact, although the synthetic network architecture is of course known, the resulting covariance structure is not known. To the best of our (and some of our more computationally-expert collaborators) knowledge, there is no general way to specify in advance the correlation structure of an attractor-style spiking network, which of course have a series of interacting nonlinearities that would make this exceedingly difficult. Regardless of this technical point, we think that direct comparison to the classical model is highly valuable. We had envisioned saving this for a second paper, as there are a plethora of potential analyses one could do to test the mechanistic models against our dataset. But we spent a lot of time on the core comparison and provide some interesting results at the end of the revised manuscript. Indeed, we were pleasantly surprised to see that the GLM did a really nice job of estimating the recurrent connectivity of the network and recovering the temporal dynamics built into the model neurons, which we take as an additional validation along the lines of the “known right answer” logic proposed by the reviews.

Third, we have also clarified the value of the simultaneous fitting (i.e., estimation of all neuron-to-neuron interactions within a recording session). In retrospect, we really goofed on making both the nature of the “simultaneous” fitting and its value clear in the first submission. The revised manuscript now clarifies how the simultaneous fitting is done (i.e., at the level of all neurons recorded within a session), and how this approach mitigates the usual interpretive ambiguities of standard pairwise correlation measures. Imagine that Cell A drives both Cell B and Cell C, without any additional drive between B and C. Standard pairwise correlations, estimated one pair at a time in series, would identify correlations not just between A and B and A and C, but also between B and C (the latter by virtue of their shared input from A, which is not considered in a pairwise correlation). But by estimating these impacts simultaneously, the fully-coupled GLM could capture the impacts of A on B and C, but can also “explain away” the correlation between B and C. This issue has been laid out nicely in Stevenson et al. (2008), which we now cite. We have also added comparisons to more traditional measures of correlations in the supplemental.

Essential revisions:1) All reviewers identified potential problems with the analysis/interpretation of the results of the GLM procedure, as can be seen in the individual reviews below. However, as detailed by reviewer 2, many of these concerns could be addressed by testing the GLM methods against synthetic data from the very network models that the results seem to support. This would require substantial work, but is in principle straightforward. This sort of analysis, where the model is tested with designed data sets for which the correct answers are known, would substantially strengthen the credibility of the results and their interpretation.

We appreciate the push to directly compare data to theory. This was planned to be a detailed second paper, and we have several collaborations underway to that effect. Nevertheless, we have made, frankly, faster and more clear progress on this than we expected. We have generated synthetic datasets from an attractor network while parametrically varying the recurrent strength of the excitatory population (Figure 7). We sampled spike trains from these networks akin to the experiment and fit the GLM in the exact same way we did for the real data. We found that coupling (estimated from the GLM) is systematically related to the internal connectivity strength of the attractor network. This finding is not only an important validation but allows us to situate our LIP and FEF results relative not just to each other, but to theory. We now conclude the manuscript with these analyses.

Some important methodological concerns that could be at least partially addressed in this way are the following:a) It is not entirely clear why the "exceptionally strong coupling" between neurons contributes in such a subtle way to the predictions of neural activity (i.e., they are not needed to accurately replicate the PSTHs).

We have clarified the dissociation between explaining the mean responses (PSTHs) and explaining the single-trial responses (which benefit from characterizations of correlated noise, both between and within neurons). This dissociation is commonly observed in such single-trial fitting schemes, such as the classic population GLM paper by Pillow et al., as well as in more recent work in LIP and MT (Park et al., 2014; Yates et al., 2017). It’s worth mentioning that the contribution of coupling to the predictions of neural activity are not “subtle” in the data. As explained, coupling has little effect on predicting the PSTHs because they collapse across trials and ignore correlations (Figure 3D), but coupling universally benefits the predictions for single-trial spike trains (Figure 3E). As noted in the manuscript and in our more detailed replies, to understand the network dynamics underlying persistent activity during behaviorally relevant timescales, we must focus on single trials rather than averages that obscure the single-trial dynamics (for more discussion see Stokes and Spaak, 2016; Latimer et al., 2015). That said, it is also important to note that the GLM does a very good job predicting the PSTH of most neurons regardless of coupling (variance explained in Figure 3D) which is also consistent with the work cited above. We have attempted to clarify and emphasize this point in the manuscript.

b) Verifying the shuffling procedure (null hypothesis) for determining the significance of the coupling terms.

This appears to reflect a major weakness in our description of our validation procedures. Either that, or we’re not exactly sure what additional verification of the shuffling procedure (already akin to a permutation test, which is a standard nonparametric validation) would be. So, we’ve tried to clarify the description of this and to emphasize the value of the shuffling at appropriate points. In our detailed reply to reviewer 2, we similarly state that there might have been a misunderstanding of the methods as originally described, as the reviewer appears to request an important validation that was provided in the original manuscript. Our utmost apologies, we have worked on the phrasing around that important part in the revised manuscript. We have also noted that this procedure was used to validate the coupling in the seminal coupled-GLM work (Pillow et al., 2008; see supplement) and many more recent uses of this analysis scheme (Runyan et al., 2017; Yates et al., 2017).

c) It is unclear whether the history-dependent kernels only capture intrinsic neuronal properties, as opposed to influences from other neurons that could be equivalent to history effects.

We have clarified this point in the revised manuscript. At a first pass, the simultaneous nature of the fit allows the model to parse out distinct effects of history and other neurons (in addition to task factors). This is another facet of the “explaining away” issue described above. However, at a deeper level, this “explaining away” is only as good as the data are complete, and so it is indeed possible that some of the history properties reflect the impacts of unobserved inputs as opposed to intrinsic cellular properties. We have now clarified this in the manuscript.

d) How do the results depend on receptive field overlap? In particular, it is possible that the differences between LIP and FEF are the consequence of different overlap patterns in the two areas.

We did not map the 2D spatial extent of the receptive fields in detail. For the two-area ensembles we recorded from, doing so would require another more complex (e.g., white noise) stimulus and additional data collection time. We would love to have such data but in the interest of statistical power and stability for the main experiment, we did coarser online mapping of the ensemble response fields. Therefore we cannot estimate the precise overlap of the receptive fields. Few studies have done this during working memory tasks, but it’s a very interesting question that we had considered previously, and our thinking was that the very large size and idiosyncratic shapes of RFs in these areas may dilute the potential for nuanced insights with the topic becoming the primary focus of an experiment.

Regardless, we did our best to explore this issue in our data. Classical experiments (and simulations) use a simplified 1D representation of the spatial selectivity: the angle (saccade direction) from fixation. We have added a supplemental figure showing the relationship between coupling and the similarity of the neurons’ spatial tuning using this type of tuning curve (Figure 6—figure supplement 1). However, critically, the differences in coupling between LIP and FEF are present even during the fixation period when there is no visual stimulus (or memory of the stimulus) driving the RFs of the neurons (Figure 6A). Therefore, this result is hard to explain simply by differences in selectivity for the stimuli. This result is in accordance with previous findings that relied on a similar analysis and logic (Katsuki et al., 2014). We have also elaborated on this topic in the Discussion.

2) The total coupling strength, which is the net result of adding the positive and negative (inhibitory) kernel weights, is used as a summary measure of coupling between neurons. As elaborated by reviewer 1, this is a potentially misleading quantity, as it does not consider the magnitudes of the excitatory and inhibitory components. The analyses should be run also with alternative measures that take this distinction into account.

Absolutely a fair point. We’d previously explored a range of metrics and now include alternatives. No major changes in the pattern or interpretation of the results present themselves. We do wish to point out that quantifying the coupling impact by calculating the sum of the kernel weights over time is indeed quite coarse, but it is not “misleading” because the magnitudes and excitatory and inhibitory components are considered in the sum. So this seems a pretty legitimate way to simply describe the time-averaged net impact. The major shortcoming of this approach is that it collapses over time, but that’s what the remaining analyses that use PCA serve to add.

3) Consideration of differences across neuronal classes; in particular, whether the results are distinctly different for neurons with significant delay period activity, as mentioned by reviewer 2.

We were interested in this as well, but unfortunately no clear patterns emerged. We have added a supplemental figure depicting what the reviewer suggests (Figure 4—figure supplement 1).

4) There is insufficient validation of the model against traditional analysis techniques, such as cross-correlation analyses.

We had performed these internally as an initial validation before feeling good about relying on the GLM fits, but we now add these in the revised manuscript and the supplement (Figure 2—figure supplement 1). However, from our perspective, these more traditional assessments of correlation suffer from a weakness that the population GLM was designed to fix (as described previously; Stevenson et al., 2008). Additionally, this GLM approach has been validated in physiological circuits/populations where the connectivity was better-known (Gerhard et al., 2013; Pillow et al., 2008). Bottom line, we were in a bit too “in our heads” in trusting the GLM, but we hope we’ve shared convergent evidence to build the reviewer’s trust in the technique now.

5) There is insufficient description of some of the analytical procedures (see specific comments below).

Agreed. Internally, we had relied on nearly-identical analysis schemes in Yates et al. (2017) and Park et al. (2014) and were likely too internally context-deep when we summarized the methods to others. We are big proponents of manuscripts standing on their own and have expanded the treatment of the analytic procedures both in the methods and at various interpretive points. These are indeed important clarifications, which are of relevance to some of the major/general issues described above. Thanks for pushing us to improve on this axis.

Reviewer #1:This work addresses important problems in oculomotor physiology, is sound, clearly described, and contains many interesting nuggets about the FEF-LIP circuitry. Just to mention one of them, the three interaction kernels provide convincing measures of the time constants of these circuits, confirming long-help ideas based on theory and modeling. Other results, such as the stronger interactions across than within areas, are interesting precisely because the do go a bit against the grain.I only have one substantial comment. Several analyses are based on the total coupling strength, which is the net result of adding the positive and negative (inhibitory) kernel weights. This leads to the conclusion that "coupling in LIP was greater than FEF, suggesting a greater degree of recurrent connectivity in LIP". But the total strength is a crude measure. If the subtraction of excitation and inhibition gives 1 = 20 – 19 net excitation in one area and 2 = 5 – 3 in the other, is it really fair to say that the coupling is stronger in the latter? Clearly not. Stronger nonlinearities are expected for highly balanced circuits in which both excitation and inhibition are stronger. This intuition is missing in the Discussion paragraphs four and five. In that case, identical excitation in both areas but stronger inhibition in FEF than LIP would be perfectly consistent with stronger winner-take-all dynamics in FEF and more tolerance to spatial ambiguity in LIP, in turn consistent with saccade selection in the former and divided attention in the latter.My suggestion is to quantify the excitatory and inhibitory interactions separately. For instance, by computing the total excitatory coupling on one hand, and the total inhibitory coupling on the other. If both terms AND the net result are stronger in LIP than FEF, then that would credibly verify the current conclusion ("greater coupling in LIP than FEF"). But I suspect (from Figure 3B) that the situation is a bit more nuanced. For instance, stronger inhibitory coupling in FEF would be consistent with stronger motor competition and/or narrower visual tuning. Likewise, one could argue that the relevant indicator of stronger or more reliable delay-period activity is not the net coupling (excitation minus inhibition) but the total absolute coupling (i.e., the total summed magnitude of the coupling weights). I think that, in general, consideration of these alternatives beyond the net coupling would provide a broader and more substantial view of the data, and a more nuanced basis for their functional interpretation, without any alterations to the GLM model.

This is a valid concern; we struggled to find a good way to summarize the coupling with a single metric that was devoid of caveats. We have plotted the absolute coupling (Figure 3—figure supplement 1A), and the primary result holds. Considering the excitatory/inhibitory comparison, in the current analysis (just to clarify) the coupling strength is only summed for each individual kernel across time, not across kernels, providing a simple scalar measure of the net impact of one neuron’s spikes on another. Thus, the histograms in Figure 3 show the net relationship of each neuron pair, and show how many neuron pairs were (in general) excitatory and inhibitory for each area interaction (the tails of the distributions). It is true that this is a coarse metric and can obscure the changes in positive and negative weights over the time period of each kernel, but we were looking for a simple descriptive measure of the overall relationship of each neuron pair. If we were to sum all positive and negative weights for all time bins separately we lose the fundamental relationships between the kernels (and thus the pairs of neurons), which can be challenging to interpret. Nevertheless, we have included separate comparisons of the overall excitatory and inhibitory interactions as well (Figure 3—figure supplement 1B,C). Most importantly, this is one of the reasons we used PCA to summarize the temporal profile of the kernels in a more principled way.

Reviewer #2:This manuscript provides evidence that recurrent mechanisms in LIP, FEF and between these two areas are engaged during persistent activity in a delayed response task. This is supported by fitting neuronal spiking data simultaneously recorded in these areas with a population encoding model. A GLM fitting procedure estimates parameters for visual and motor dependencies, neuronal history dependence, and coupling kernels between simultaneously recorded neurons. The coupling kernels are reported strong in comparison to history kernels, and contain time scales consistent with excitatory and inhibitory mechanisms typically included in attractor models of working memory.The manuscript is a valuable and significant contribution to the research on the mechanisms of working memory. For this classic task, a population-level analysis of simultaneous recordings in two cortical areas has not yet been presented, and provides important insights into the relative importance of different mechanisms that have been proposed to support persistent activity: intrinsic mechanisms, local circuit recurrence, and recurrent interactions between areas. The methods applied are aimed at testing this with advanced statistical methods.My major concerns regarding the conclusions drawn from these analyses are:1) There is an inconsistency in the manuscript in that, on the one hand, interneuronal couplings do not improve PSTH predictions (Figure 3D) and yet various analyses (Figure 3A, Figure 4C) are interpreted as demonstrating "exceptionally strong coupling", "exceptionally strong recurrent interactions" or that "network dynamics play a more vital role than intrinsic dynamics". If recurrent interactions are removed from an attractor model persistent activity collapses, and PSTHs are profoundly affected. At face value, the findings thus do not seem to represent a direct support of these models. The fact that exceptionally strong couplings do not have an impact in the PSTH should be better explained.

This is an important issue and (as stated in the general reply above) one we have tried to clarify substantially. PSTHs only quantify the average responses of the neurons across trials. The memory of the target itself, and thus, the neural dynamics act on single trials, the behavioral relevant time periods. Our findings show that coupling (and thus the correlations between the neurons) is not necessary to capture the average responses, (i.e. considering the spike trains from each neuron as independent would give you the same result; Figure 3D). But, if you look at the predicted spike trains on individual trials (recalling that the model was fit on individual trials), you can see that there is useful information in the correlations between neurons that increase the accuracy of predictions of single trial responses (Figure 3E). Hence, the interesting result here is that the correlation structure matters for the population code on individual trials. It is not surprising that it does not matter for the average responses across trials because PSTHs obscure (i.e. average away) single-trial fluctuations by design. This distinction was also found in the retina in the seminal work applying the coupled-GLM approach to modeling population data (Pillow et al., 2008). This finding emphasizes that PSTHs may be a limited way of trying to understand the population code on the time scale that matters for working memory. We have attempted to clarify and elaborate these points in the revised manuscript.

2) I am concerned about the "null hypothesis" in Figure 3A: if trials are shuffled for each neuron before the fit, then neural activity is modeled with spiking data of other neurons obtained in trials with different delay duration, so not aligned to the stimulus and saccade predictors used in the model. Since these events are typically associated with an increase in firing rate, interneuronal couplings will be reduced in this model because of these misalignments. The reading of "strong recurrence" that now emanates from the comparison with this "null hypothesis" might instead result from this misalignment. Maybe a more direct assessment of how strong is the contribution of couplings to the network would be to evaluate what fraction they represent in the sum of all predictor terms in the model.

Thanks for raising this! We believe there is an essential clarification (but not a problem with the shuffling procedure). As noted in the manuscript, the timings of the trial events and spikes within each trial are always preserved in the GLM analysis. Only the order of the trials is shuffled (i.e., the trial “labels”), e.g. what might be considered trial 1 for neuron 1 is trial 23 for neuron 2 etc (but each neuron still gets the “right” event times for that spike train within each trial). Therefore, only the temporal relationship between neurons is disrupted. That is why this approach is an especially strong test of the presence of coupling, because only the correlations between the neurons are eliminated while the task dependence and autocorrelation are unaffected. This procedure was used to validate the presence of coupling in the seminal neural GLM paper (Pillow et al., 2008).

With that clarification, we hope it’s clear that the coupling predictive index (Figure 4) is in essence what the reviewer suggests. It describes the contribution of the coupling filters in terms of their deviance explained (a classic goodness-of-fit measure for GLMs) as a fraction of the deviance explained by all the predictors in the ‘“full” model (Materials and methods). Sorry, again, for not being sufficiently clear about this.

3) The manuscript emphasizes the role of recurrent interactions in generating persistent activity. However, not all neurons recorded in LIP and FEF show persistent activity in this task. In particular, Figure 2 shows that neurons with robust persistent activity show sustained history kernels, which supposedly affect just a minority of neurons in the database (Figure 4C). Some of the conclusions may change if one considers separately neurons with strong persistent activity from other neurons. For instance, if neurons with persistent activity mostly populate the right cloud of dots in Figure 4C the interpretation of this panel's results would be very different. Example neurons shown in Figure 1 could be explicitly identified in Figure 4C and other panels with individual neuron data. If the question is how persistent activity is generated, then it should be relevant to consider separately neurons with and without persistent activity.

These were our thoughts exactly. We conducted these analyses, but unfortunately there was no systematic relationship between traditionally determined persistent activity (from PSTHs) and the coupling or history terms. This is demonstrated in the supplement by an updated version of Figure 4C (Figure 4—figure supplement 1). We believe this finding reemphasizes the above point – that single-trial dynamics and the across-trial average responses may be very different, and the traditional ways of describing single-neuron activity may be too limited to capture the dynamics of the network during which working memory is actually active. Furthermore, we acquired large somewhat unbiased samples of neurons from the ROIs, and although we looked for neurons with traditionally-determined persistent activity to place our recording probes, we included all isolated units in our analyses. Our goal was to get a broad sample of the network. Therefore, unlike classical experiments, we do not cater the stimulus to a single neuron’s RF (this would be impossible, as we get many neurons with a variety of RFs). Hence, this difference in approach may have an effect on the traditional measures of persistent activity in the single neuron PSTHs. As approaches have moved to large scale simultaneous recordings, we must revisit how we go about analyzing and thinking about this kind of data. This topic has been expanded in the Discussion.

4) The interpretations in the manuscript jump very easily between the statistical description contained in the fitted kernels and biophysical mechanisms, and this should be more carefully considered. Any "temporal integration above and beyond that explained by the external task elements" (Park et al., 2014) and the spike times of the few simultaneously recorded neurons in that session will be attributed to history kernels, which would therefore be inappropriate to attribute to intrinsic cellular mechanisms. In general, authors should be cautious with the possible confusion in using "intrinsic" to refer to the history kernel. The history kernel might include traces of other network dynamics not captured by the few neurons recorded simultaneously in the session.Importantly, many of these concerns could be directly addressed by analyzing spike trains from a few neurons selected from attractor network simulations (many of which freely available) to verify if their GLM approach is able to extract the biophysically-based parameters of the model, as currently argued. My intuition is that model neurons without any specific intrinsic mechanism with long time course would show a long-lasting history dependence (self excitation) when modeled against the activity of just a handful of their close neighbors. The number of neurons used as predictors, or the similarity of their memory fields, could also conceivably have an impact in the decay time of history and coupling kernels, and thus contribute to possible differences between FEF and LIP in this analysis. These computational models are freely available (for instance https://neuronaldynamics-exercises.readthedocs.io/en/latest/) and the model data can be generated easily and analyzed with the authors' methods.

Thanks a million! This is a good point and we are deeply grateful for the pointer to that model.

The reviewer has a valid concern regarding over-attributing the history effects to intrinsic biophysical mechanisms. It is true that the model cannot account for the influence of unobserved neurons (which no doubt play a substantial role), although to be fair, all electrophysiological recordings and analyses deal with this limitation. However, only the fitted neuron’s own spikes impact the history kernel, and although the model cannot account for the role of unobserved neurons driving that spiking, it is not true that simply any unexplained variance by the other predictors will show up in the history kernel (i.e., they would need to be well correlated with the cell’s spike train). All parameters are fit in the model simultaneously, and only the predictive power of the neuron’s own spikes can contribute to the history term. The history kernel does a good job of capturing the neuron’s autocorrelation while partialling out the effects of the other (observed) predictors, and hence gives a reasonable estimate of “self-drive,” but it’s true we cannot make any specific claims about the biophysical mechanisms, only statistical relationships. We have edited the main text to remove and clarify the usage of intrinsic in reference to the history kernels. It’s also worth mentioning that GLM approaches can extract accurate parameter estimates of intracellular mechanisms when modeling these explicitly (Latimer et al., 2014).

We thank the reviewer for suggesting this resource for the spiking attractor network models. We have used it for our additional analyses that conclude the manuscript (Figure 7). We highlight several important insights (above and in the manuscript), but of relevance here: the analysis of the model recovers the temporal dynamics of the neurons with known properties, so this is a useful validation.

Reviewer #3:The paper uses a novel analytical technique to infer functional connectivity among FEF and LIP cells during a memory guided saccade task. The analysis takes a GLM-based approach and fits spike trains of simultaneously recorded FEF and LIP cells as weighted combinations of kernels related to (1) target location, (2) saccade direction, (3) coupling filters modeling interactions among cells and (4) history filter modeling within-neuron autocorrelations. The main conclusions are that FEF and LIP neurons show strong coupling both within and across areas, and that LIP shows stronger coupling and longer intrinsic timescales than FEF.While the approach is interesting and potentially useful, I am afraid it is not sufficiently justified in the present version of the manuscript. The level of detail provided is simply not sufficient to support the claims based on this novel analysis technique. I see two main problems with the manuscript as it is now:1) There is insufficient control for potentially differential sampling of neurons in FEF and LIP: Could the apparent stronger coupling in LIP vs. FEF be due to different neural sampling in the two areas? For instance, if the RF of the recorded cells had more overlap in LIP relative to FEF, that may produce stronger apparent coupling in two different ways. Stronger coupling may arise if the model mis-attributes shared stimulus/saccade related activity to the coupling weights. The authors emphasize that their model identified coupling above and beyond any stimulus-related response, but they provide little evidence to substantiate this assertion. Stronger coupling may also arise if the model correctly attributes variance but the neurons are in fact more coupled by virtue of being more functionally similar. In that case, the areal differences are only artefacts of the neurons the authors happened to sample. We need much more thorough analyses of the similarity in RF profiles and temporal response profiles of the recorded cells to rule out a sampling confound.

This is an important point. The differences in RF organization between these areas and the implications for network architecture is an interesting (and open) question, but unfortunately our dataset (as with most prior work) is not equipped to perfectly disentangle sampling noise from true areal differences. To really nail this issue, we would need to more precisely map the spatial extent and temporal profile of very many neurons with a dynamic (white noise) stimulus in addition to the current experiment (and/or to already know the full distribution of RFs in each area).

It’s worth mentioning that the 2D nature of the receptive fields (and the memory representation) is often simplified in classical experiments (and simulations) to 1D tuning curves representing the saccade direction angle from fixation. In this classical form of the analysis, the spatial structure and extent of the RFs are summarized rather coarsely. Despite this missing detail, we compared the coupling between neurons as a function of their spatial tuning similarity in an effort to address these questions (Figure 6—figure supplement 1). For all area interactions coupling was slightly stronger for pairs of neurons with more similar spatial tuning, and that relationship was most pronounced for neurons within LIP, a feature predicted by classic attractor network models. However, most importantly, the overall differences in coupling between LIP and FEF were also present during fixation when there is no visual stimulus in the receptive fields or memory being maintained (Figure 6A), meaning the coupling differences could not be simply due to differentially driving RFs with different degrees of overlap. This result points to differences in the functional connectivity.

Of course, differences in sampling can always be a potential problem in electrophysiology experiments, but we conducted the experiments using identical recording equipment, under identical conditions within the same sessions (which we suggest is the best we could do given current methods viable in these areas in the macaque). If we may muse a bit more beyond the pragmatic realities of currently available data, it’s intriguing to appreciate that functional differences in the organization of receptive fields may reflect differences in the underlying architecture; there is no strong reason to assume identical coverage of the visual fields with respect to receptive field sampling. Thus, differences in coupling due to receptive fields could be due to the underlying cortical organization and potentially interesting rather than just an artifact of sampling.

Regarding the concern that: “Stronger coupling may arise if the model mis-attributes shared stimulus/saccade related activity to the coupling weights. The authors emphasize that their model identified coupling above and beyond any stimulus-related response, but they provide little evidence to substantiate this assertion.” This is a separate issue, and as we have clarified in the revised paper (and in several of the replies above): All the parameters are fit simultaneously, so variance explained by one of the covariates can not be explained by another. In other words, the coupling kernels only explain variance not explained by the stimulus, saccade, and history kernels. So it’s not just catch all; it’s predicated on the likelihood of one neuron spiking given that another neuron spiked, given the likelihood of all other covariates in the model (effectively “partialling out” or “controlling for” their impact).

2) There is insufficient validation of the model against traditional analysis techniques. The field has several traditional, established measures of neural coupling or connectivity, including noise correlations and joint PSTHs or autocorrelograms. The authors dismiss those analyses off hand with little justification except for saying that they do not adequately remove stimulus-evoked activity. But there are actually many ways for them to get around confounds with these methods. Comparing their model results to these traditional analyses is essential to validate their results.

We are very happy to add in these analyses, as we’d actually file-drawered them after doing many such validations early in the project. We have added supplemental figures showing cross-correlograms and the relationship between correlations and coupling estimated from the GLM (Figure 2—figure supplement 1). These methods have also been validated previously with traditional correlations in physiological circuits with better-known connectivity (Gerhard et al., 2013; Pillow et al., 2008). In addition, previous work has shown the merits of GLM approaches over traditional measures of cross-correlations (see above replies for a treatment of this issue; for a previous review article on the topic, see Stevenson et al., 2008). We highlight (now, more clearly) that we jointly fit all simultaneously recorded neurons from each session, unlike correlation measures that only consider pairwise combinations in isolation. This approach allows us to “explain away” potentially misleading correlations and better estimate the connectivity of the network.

3) There is insufficient description of the analytical procedures:– How were the neurons combined across recording days? Clearly, the neurons that were simultaneously recorded each day could be fit using the described method. But the authors say that they simultaneously fit the entire data set – including neurons recorded on different days. How precisely did they set this up and how did this affect the results?

We regret the lack of clarity and have updated the manuscript. Neurons were not combined across recording days. Each daily session (including all simultaneously recorded neurons) was fit with the model. The data (and fits) were then summarized across all sessions. These points are now clarified in the Materials and methods.

– What were the kernels that the authors used in the model fits? How were those kernels selected? What properties must the kernels have to make the model work? Are the conclusions robust over reasonable changes in kernel shape?

Kernels were constructed from smooth temporal basis functions (nonlinearly time-scaled raised cosine functions) as described in previous work (Pillow et al., 2008; Park et al., 2014; Yates et al., 2017). Kernel fitting procedures were consistent with these previous studies and fits have been shown to be insensitive to reasonable changes in the lengths and shapes of the kernels. (But still sorry for not putting these details into the initial submission!). These parameterizations are conventional in the literature and simply used to capture smooth temporal dependencies and reduce the dimensionality (i.e. as an alternative to specifying a parameter at every single time bin). The Materials and methods have been expanded to include more details about the kernels to accompany the references.

– Goodness of fit and model comparisons: The authors should provide a complete description of the type of models they compared.

Of course. The description of model comparisons has been elaborated in the Materials and methods.

Figure 3D shows the goodness-of-fit as the variance explained of each neuron for the coupled and uncoupled models. Figure 3E shows the accuracy of single-trial spike train predictions. Figure 4 shows the results of model comparisons for models with and without coupling (coupling predictive index), and with and without history (history predictive index). These are measures of the deviance explained of the GLMs (Agresti, 2002).

o Individual monkey results: How many cells were recorded in each monkey? Did the critical results hold in both monkeys?

The results held for both monkeys (number of neurons: Monkey A = 459, Monkey L = 508). Symbols indicating from which subject the data came has been added to the goodness-of-fit plots (Figure 3D,E). This has been updated in the manuscript.

[Editors' note: further revisions were suggested prior to acceptance, as described below.]

Essential revisions:

The main concern communicated in the last set of reviews had to do with, not just a lack of clarity regarding the GLM validation via permutation, but a pair of regrettable misunderstandings. One to do with the shuffling, and one to do with the role of the uncoupled model. We do our best, in words and in code, to clarify these issues.

We now call the shuffling test a “permutation” process in the manuscript to hopefully avoid any misinterpretations that might be associated with “shuffling” in related contexts. That said, the permutation test is not the key statistical test – the comparison to the uncoupled model is (i.e., it supports principled validations of the GLM and model comparisons, the results of which suggest highly significant coupling contributions). The permutation/shuffling results are useful supplemental analyses because they provide reasonable estimates of null distributions in the same language and units as the main results (i.e. coupling strengths). Unsurprisingly, the shuffling analysis provides similar results to the uncoupled model, because they are simply different ways of making the neurons independent. In the uncoupled model, the neurons are truly independent (there are no connections between them) and in the shuffling case the correlation structure is disrupted which effectively makes the connections random. However, the shuffling analysis has the added benefit of providing a comparison using the full model (with all the same parameters). The field usually uses both, as we have.

We hope to have now sufficiently explained both to reviewers and to a potentially broad readership that the major criticism/concern put forth does not apply to our analysis (or to the large number of preceding papers that have used the same permutation process).

1) What exactly is the uncoupled model?

The uncoupled model does not include any coupling kernels – only task and history kernels (i.e. there are no connections between the neurons in the uncoupled model; each fitted neuron does not take into account spikes from other neurons, but it does still take its own spikes into account via its history kernel). This “dropping of terms” is akin to standard linear regression techniques (e.g., stepwise, nested tests, etc.), where terms might be added or removed and the resulting changes in quality of fit are assessed and interpreted.

The uncoupled model is therefore the standard principled statistical technique for validating the explanatory power of the coupling kernels. This is reflected in several ways in the paper: as the difference in the variance explained (goodness-of-fit) between the two models (in the PSTHs, Figure 3D and on single trials, Figure 3E), and also in the model comparisons (measured as the deviance explained between models with and without coupling and history terms; Figure 4; Materials and methods). All of these methods are used to quantify the explanatory power of adding coupling (or history) to the model over and above the other covariates. We have clarified the importance of the uncoupled model’s role in the manuscript.

2) How exactly is the shuffling across trials performed?

We very much regret that our ways of describing the shuffling procedure clearly weren’t working for some of the audience. In the last round of reviews, there was still a misunderstanding regarding how the shuffling procedure is performed – something we certainly need to get right. We deeply appreciate the reviewer’s willingness to dig in and build a simulation. Here we explain how the simulation helped us understand the disconnect here – for all readers, we wish to state clearly that the concern reflected in the simulations does not apply to our actual analysis scheme.

The shuffle-trained GLM includes all parameters of the full model (task, history, and coupling). The permutation procedure we use only disrupts the relationship between neurons (i.e., the coupling kernels) but *leaves the dependence on the task (stimulus, saccade etc.) and history for all neurons intact* (more on this below). The stimulus/trial event times and spike times are never misaligned for any spike train, and the time bins in which these events occur are never shuffled within trials (which seems to make our analysis different from what the reviewer so patiently and constructively coded up— the colored raster figure was key, thanks!). How is this done? It is difficult to conceive in the mindset of standard shuffling analyses, so let’s start fresh…

Consider Neuron A, which is the neuron being fitted in this run of the GLM. A standard fully-coupled GLM would then model neuron A’s response as a function of task events, its own spike history, and the impacts of Neurons B through Z (in the hypothetical case of 26 simultaneously-recorded neurons). The “impacts” of Neurons B through Z are simultaneously modeled with respect to the task events and history of the fitted neuron, so only the residual spikes of Neuron A are predicted by the spikes from the other neurons. The effective use of residuals to capture coupling is one of the most powerful aspects of a GLM, as it allows for the task terms to remove the signal drive, while still allowing for the unexplained activity to be used for other aspects of the fit (e.g., for the history and coupling terms). (We discuss more of these big picture issues later, but now get back to the permutation analysis implementation).

By relying on the GLM’s modeling of the task variables (which always use the correct timings while fitting each trial), the permutation can now be done. For each trial, we keep Neuron A’s original spike train (and its stimulus drive and autocorrelation is simultaneously partialed out), and we use the spike trains for Neurons B through Z from other independently/randomly selected trials (i.e., shuffled without replacement) as predictors. We do *not* then need to pretend that those spike trains came from the original trial – the goal is to solely eliminate the relationship between the neurons. The spike trains from other trials should not have any predictive power with respect to the fitted neuron’s spike train (Neuron A) beyond simple chance, allowing us to estimate a baseline level of coupling due to noise. The spike trains from Neuron’s B-Z are then analyzed with respect to the task events from the respective/corresponding trials when those neurons’ spike trains are fit.

This scheme effectively treats each neuron *independently* (analogous to removing the simultaneous nature of the recordings and is similar to the uncoupled model where the neurons are independent by design, i.e. there are no coupling kernels). But, again, the single-trial time courses of the spikes and task events for each neuron are *unchanged*. So, from the perspective of a fitted single neuron, nothing is different except its relationship to other neurons in the session; the fitted neuron’s autocorrelation and stimulus dependence is preserved. This can be seen in the PSTHs, which are *unaffected by this shuffling procedure* – and shouldn’t be because the model’s fitting of the task event terms is still totally proper. In summary, this model-based analysis is flexible enough to use trials with different timings (because it is fit on single trials) to allow the signal correlations to be maintained while the noise correlations are disrupted.

In practice, of course, no model is perfect. The possibility of mis-modeled aspects of the neurons’ responses is certainly possible, but because the task events are randomly-timed from trial to trial, this is solely likely to create coupling estimates around zero simply due to noise, as opposed to having any bias (which can be confirmed via simulation of neurons that are independent by design). Additionally (and critically), since the task events are jittered trial by trial, any “bleed through” effect of task dependence on the coupling terms from other trials actually makes the shuffled comparison a *more* conservative estimate, because this would artificially inflate the null distribution and hence make our estimate of the true coupling smaller.

Although the scheme is clearly not straightforward to explain, especially in the historical context of standard CCGs and shuffling controls, we hope that it is not just clear, but more directly addresses (i.e., models out) the potential for stimulus drive contamination, which many clever but imperfect traditional shuffling controls attempt to sidestep. But it’s nothing magic – the catch is that if you want to use a GLM, you need event timings to be randomized from trial to trial to decorrelate the predictors (and you can use all trials for the coupling analysis); if you want to use CCGs, you prefer to have multiple repetitions of trials that are as identical as possible. We favored the former technique because it allows for all simultaneously recorded neurons (B-Z) to go into the coupling for Neuron A at the same time, thus being able to “explain away” partial correlations in a way that standard pairwise correlation analyses cannot (the GLM also has other nice properties, because it provides descriptions of a cell’s own spike history and responses to various task events, all in the same fit and language). Finally, the permutation analysis is really just meant to be an intuitive way to visualize the magnitude of the observed coupling relative to a reasonable null model. The statistical comparison between the fully-coupled GLM and the uncoupled model (i.e., the GLM without the coupling terms) is the more straightforward test of whether coupling adds to the quality of fit and is also used when assessing the relative impacts of coupling and history via the model comparisons (between models with and without coupling and history).

We have clarified this in the body of the manuscript and in the Materials and methods. In addition, this is how such a model has been described in previous work, both originally and more recently (e.g., Pillow et al., 2008; Yates et al., 2017; Runyan et al., 2017). Runyan et al. shows a more recent example of this shuffling control for the GLM which was conducted (and explained) in the same way (Runyan et al., 2017).

In addition, we have attached some code and figures demonstrating how the shuffling/permutation procedure works with an example session looking using cross-correlations and have also shared a simplified version of the GLM fit to simulated data from the attractor network. Jonathan Pillow also has a series of detailed GLM tutorials using the same sort of approach on his website which may be beneficial: https://github.com/pillowlab/GLMspiketraintutorial.

Again, we really appreciate the reviewer putting the time into the code and figures. To revisit the clarification in the context of their generous analysis, in the reviewer’s example (in the non-shuffled figure), imagine shuffling only the rows (the neurons’ identities) within each panel (each trial) – the stimulus alignments (and spike times) over the time course of the trial would remain unchanged. If you do this for each trial, the relationship between neurons across trials is broken but the within trial time courses are preserved. This is not exactly what we did, but is similar conceptually to what we explained earlier. Instead, in our data structure, each panel would be a neuron, the number of rows would be trials, and the columns would still be time bins (i.e. conventional raster plots for each neuron). Now, if we shuffle the rows (the trials) for each other neuron (each panel) independently, nothing has changed for the fitted neuron whose response we are modelling, the single trial timings are the same, and the PSTH (which averages across rows in this scenario) is identical, but now the relationship to the other neurons is made random. Thus, the true timings of the covariates in the GLM (the stimulus on, the saccade, spike times etc.) on single trials are always used. Keep in mind that, because in our task (unlike traditional ones) the trial parameters are variable on every trial, the analyses must always be referenced to the values on individual trials (e.g. PSTHs are plotted aligned to the target or the saccade; spikes are counted in windows around those events on each trial), so single-trial timings must always be preserved. In addition, that means it’s impossible to construct some sort of conditioned-matched surrogate dataset with the shuffling analysis since every trial is unique.

In the reviewer’s code, it appears that a difference might arise because the randomization is done on every trial with replacement and thus there can be an effect of “double dipping,” but it’s hard to know for sure without being able to run the code (we tried; more on this below). To replicate our permutation procedure in spirit, one would want to do the permutation in the neuron loop, or make sure the trials are sampled without replacement. In essence, the reviewer has also highlighted why cross-correlations are fraught measures for estimating connectivity – by showing how illusory correlations can arise between independently simulated neurons. This reiterates one of the points of using the permutation procedure as a control, because one can actually estimate the amount of coupling due to chance alone (in effectively independent neurons) and use that to construct a reasonable null distribution.

3) More careful analyses of the activity during fixation and early delay.

Please see below for detailed replies – we have tried to clarify these issues in the manuscript, and remove any implications/interpretations for other models that were deemed overly ambitious.

4) Clarification of the dependence on overall firing rate (separate from the variations in connectivity).

We believe this issue also stems from the regrettable mutual disconnect regarding the GLM and the shuffling procedure. One of the benefits to using the GLM approach is to avoid confounds like this, and give a better estimate of interneuronal interactions (i.e, more pure, via “modeling out” the task events). Baseline firing rates are captured by an additive (constant) term in the GLM. Coupling can be thought of as lagged correlations in the residuals after controlling for the other covariates (i.e. visual/motor responses, auto-correlation, and baseline firing rate). Thus, estimates of coupling already partial out effects of baseline firing rate.

To further assuage any remaining wonder about whether the overall differences in firing rates between the areas could affect coupling, the shuffling control comes in handy again. The coupling kernels for each area were compared to null (shuffled) distributions within each respective area, hence the spike rates in the normal case and the shuffled control were identical (the number and time of spikes are unchanged), but during shuffling the relationship between the neurons is obliterated giving us an estimate of coupling simply due to noise. Therefore, we have an estimation of how much coupling exceeds what is expected by chance for each particular area while controlling for differences in spike rate. We have included an additional supplemental figure demonstrating that this holds in the GLM analysis of simulated data from the attractor network (Figure 7—figure supplement 1). Furthermore, we have now included data on the mean firing rates in LIP and FEF, and FEF actually has significantly higher mean firing rates than LIP whereas LIP had much stronger coupling. Thus, this provides empirical confirmation that coupling is unlikely driven by a confound in firing rates.

These points are discussed at length by reviewer 2 below.Reviewer #2:The revised manuscript contains significant improvements over the previous version, notably the new computational model data that provides support to the methods applied. However, I still maintain some relevant concerns on the controls required to support the interpretations of their analyses as couplings between neurons. This concerns primarily the shuffle approach and the control of firing rate confounds as detailed below.1) Related to previous Major concern 1.I realize that part of the confusion here is that the “uncoupled model” is never fully described. I understood that this was the full fitted model, to which couplings were set to zero. From this model I would expect PSTHs to differ largely from measured PSTHs, if couplings are contributing significantly to the GLM. Instead, I gather from (Pillow et al., 2008) that the uncoupled model is a reduced model refitted to the data, and then I understand that the fit may be good even if couplings for the full model are strong contributors to the fit. I urge the authors to explain clearly what their uncoupled model is. In fact, in the current revision this is further confused in the Materials and methods (“Unsurprisingly,…”), where the “uncoupled method” is commented in the middle of the description of the shuffle approach. These two separate controls should be clearly distinguished and specified so the reader understands their separate implications. Now this is very confusing.

We have revised the manuscript to make this more clear. Thank you for calling our attention to it. In addition, as mentioned previously, the inclusion or not of coupling parameters in the model or their disruption with shuffling has little (or no) effect on the PSTHs because these are averages across trials that by their very nature obscure the trial-by-trial correlations between neurons (Figure 3D).

2) related to previous Major concern 2.I am still not convinced about the shuffle approach on which the "extraordinarily strong coupling" result is based. If I understand it correctly, the structure of the data is as schematically depicted in Decision letter image 1, with simultaneous spike trains recorded (here 10 neurons in 10 lines per panel) for separate trials (here 8 trials in 8 panels), where in each trial the stimulus is presented in different locations (here different colors of the simulus and response shadings) and delay durations are also different from trial to trial. Neurons respond with spiking activity depending on whether the stimulus falls within their receptive field or not.The GLM for the neuron on the gray background would then model spikes based on its specific stimulus and response events for each different trial, on its previous spikes in the corresponding trial, and on the previous spikes of all other neurons in the corresponding trials. Crucially the spikes for these other same-trial neurons are collected based on the same stimulus location, response and delay duration.If I understand correctly the shuffle condition applied in Figure 3, the GLM now models the gray-background neuron based on its specific stimulus and response events for each different trial and on its previous spikes in the corresponding trial, exactly as before, but now the spikes of other neurons are taken from randomly picked trials. This is represented in the schematic in Decision letter image 2 (notice the first lines of each panel are the same for both schematics). If this is the case, then now the spikes for these other shuffled neurons were not collected based on the same stimulus location, response and delay duration as the modeled neuron. As a result, the consistency of stimulus activation of neuron 9 and neuron 7 shown in Decision letter image 1 (they are both active or inactive in response to stimuli in all trials), and the consistency of the timing of response events, are broken with the shuffle in Decision letter image 2. This will lead to a radical change in the cross-correlations between these two neurons, even if these spike trains were totally independently generated.<Decision letter image 2>I have run this simulation. Data for these spike trains were independently generated for each neuron (so with no coupling between them), only keeping the consistency of stimulus, response and delay within trials. As shown in the histograms in Decision letter image 3, the mean cross- correlations (as a proxy of the couplings obtained with the GLM) were radically affected by the shuffling procedure. This does not demonstrate any functional coupling between the neurons, but the fact that task events that elicited neural responses were misaligned by the shuffling procedure.In my view, this effect could explain the extraordinary changes in shuffle histograms in Figure 3. To clarify this, the shuffle should be performed only between trials with equal stimulus location and delay duration, or else apply other commonly used resampling procedures to estimate cross- correlations (Fujisawa et al., 2008; Katsuki et al., 2014), such as within-trial jitter correction methods (Amarasingham et al., J Neurophysiol, 2012).

Unfortunately, we were unable to run the attached simulation code. There were errors in some of the lines and the intention of the code at this point was a bit unclear to us, so after a few attempts we ended up reading the code and studying the figures – regardless, they were very helpful! We hope that the main issues have been explained to the reviewer’s satisfaction in the earlier reply. We have provided examples of how the shuffling procedure works using real data and simulations in an example session with cross-correlations and also with the GLM. This illustrates what we have discussed in the general reply above. The confusion comes from a simple misunderstanding of how we permute the data, and how the GLM preserves the task dependence, which we hope is now clear. We have also added additional supplementary figures showing the shuffling analysis of the simulated attractor network data and the possible firing rate confound (Figure 7—figure supplement 1A,B). See below for more details on the possible firing rate confound.

3) The section on “Interneuronal coupling is dynamic across behavioral epochs” is very confusing and is not really addressing the point that they are formulating. The fact that there are not very marked changes between couplings measured in the fixation and delay periods is taken as evidence of dynamical working memory without much justification. Later on the comparison of early and late delay periods yields stronger couplings as delay progresses. In my view, this attempt to contribute to the debate on dynamic delay activity, or even activity-silent working memory (Stokes et al., 2015) is overambitious and needs more specific analyses (consider for instance possible traces of the stimuli in the couplings after the response period, by comparing in and out conditions in the baseline period where there is no confound of firing rate for in/out conditions, e.g. Spaak et al. SfN 2017 #339.14; Barbosa et al., Biorxiv, 2019, doi:10.1101/763938). Further, in the interpretations of this paragraph it is disclosed that their measure of coupling is confounded by changes in firing rate (“This finding highlights the potential role of nuanced changes in the dynamics of the network rather than an overall increase in population activity”). This should be declared early on, as this confound may affect many of the interpretations. Any interpretations of difference in coupling strengths should be cautious in previously demonstrating no difference in firing rates, or else in applying more stringent rate-correcting methods such as jitter correction (Fujisawa et al., 2008). Currently, the manuscript does not parallel each of the coupling comparisons (FEF-LIP, fixation-delay, etc.) with corresponding comparisons of mean firing rates, or rate-correcting measures, to support these conclusions.

Good points. It was not our intention to weigh in definitively on the debates about dynamic or stable coding, or activity-silent working memory. We are simply reporting what we found from analyzing the network dynamics during different trial epochs, which (like several other aspects of these data) will be of interest to these debates, but are in this case just the first descriptive step. We have attempted to remove any interpretations or implications that could be viewed as overreaching, as we believe a complete treatment of these issues warrants a rather long follow-up study as well. We do note that dynamics in the population activity (which are commonly observed) do not necessarily mean a dynamic code for working memory (Murray et al., 2017), which is definitely not what we wished to imply. We have rewritten the text to be more responsible and constrained here.

We also hope that the earlier responses explain why the GLM coupling estimates are inherently “rate-corrected.” We also note that the GLM parcels out the effect of the stimulus – both “in RF” and “out of RF” covariates are fit simultaneously, thus, the model is the same in both conditions (in the different trial epochs), as are the stimulus and task parameters.

Regarding: “Further, in the interpretations of this paragraph it is disclosed that their measure of coupling is confounded by changes in firing rate (“This finding highlights the potential role of nuanced changes in the dynamics of the network rather than an overall increase in population activity”)”

We were puzzled why this statement in the conclusions might be taken as admission of a flaw regarding a firing rate confound in the GLM analysis. Any help would be appreciated here. We re-read it a few times ourselves, trying to figure out how to avoid this unintended interpretation, but in the context we think it is clear that statement is simply highlighting the connection to other work (Murray et al., 2017, see above) that calls attention to the possibility of a more complicated story in the dynamics rather than a simple overall increase in persistent activity. (We discuss the firing rate issue both above and below.)

4) The computational model is a nice demonstration of their methods, in particular of the retrieval of the correct time scale of the interactions between neurons (the AMPA and NMDA current time courses). However, in terms of the strengths of the couplings, the model may also be confounded by changes in firing rate. It will be useful to plot the mean firing rate of model neurons as recurrent connectivity is increased. If this mean firing rate is increasing too, then some compensatory change may be introduced to maintain rates constant as recurrent connectivity increases in order to resolve this concern. The stimulus kernels derived from the model in Figure 7D, left panel, do not seem to correspond to the simulation results: stimuli in receptive fields should generate increase in rates, and stimuli outside the receptive field a suppression of rates.

As discussed in the response to general questions above, the GLM analysis parcels out changes in firing rate, and we also have provided an additional supplementary figure addressing this concern using the simulated data from the attractor network (Figure 7—figure supplement 1). This figure shows that coupling estimated from the GLM provides an accurate inference of the recurrent connectivity of the network regardless of changes in the mean spike rates. This analysis is yet another sanity check showing that the shuffling procedure disrupts the estimate of coupling of the network without affecting the time course of the neurons (or their PSTHs) and maintains identical spike rates as the true condition.

Specifically regarding: “The stimulus kernels derived from the model in Figure 7D, left panel, do not seem to correspond to the simulation results: stimuli in receptive fields should generate increase in rates, and stimuli outside the receptive field a suppression of rates.”

Thanks for the careful look at this, we checked the analysis and simulation and believe everything is OK here. The large bump (in the “in RF” condition) captures the transient response to the stimulus, the rest of the time course is simply noise around zero (a gain of 1) in both conditions. We are only sampling from a small number of neurons, and if the stimulus is not presented in their RFs we should expect no changes in their responses in either direction (which is what you see in real data for such conditions). Even in the idealized scenario in the attractor model, a suppressive effect would depend on how far the stimulus is presented from the RFs of the neurons you are observing, the strength and width (in RF size) of the stimulus, and the strength and range of the inhibitory connections. Thus, there are many potential factors at play here.

5) Because of the possible rate confound that affects the coupling measure, the manuscript should provide a mean firing rate description of the various areas and task periods, for instance through average PSTHs.

We have discussed the possible firing rate confound in the other responses, and now we have included descriptions of the mean firing rates of both areas in the Materials and methods. Neurons in FEF actually had a significantly higher mean firing rate than LIP (FEF: 36 sp/s; LIP: 28 sp/s; P=1.16x10^-5^Wilcoxon Rank-Sum test) which provides further evidence that coupling was not artificially inflated by differences in firing rate given that LIP had much higher coupling the FEF.